



# The asymmetric geospace as displayed during the geomagnetic storm on August 17, 2001

Nikolai Østgaard[1], Jone P. Reistad[1], Paul Tenfjord[1], Karl M. Laundal[1], Theresa Rexer[2], Stein
E. Haaland[3,1], Kristian Snekvik[1], Michael Hesse[1], Stephen E. Milan[4,1], and Aanders Ohma[1]

[1]Birkeland Centre for Space Science, Department of Physics and Technology, University of Bergen, Norway
[2]University of Tromsø, Norway
[3]Max-Planck Institute, Göttingen, Germany
[4]Department of Physics and Astronomy, University of Leicester, UK

**Correspondence:** N. Østgaard (nikolai.ostgaard@uib.no)

**Abstract.**

Previous studies have shown that conjugate auroral features are displaced in the two hemispheres when the interplanetary magnetic field (IMF) has a transverse (Y) component. It has also been shown that a $B_Y$ component is induced in the closed magnetosphere due to the asymmetric loading of magnetic flux in the lobes following asymmetric dayside reconnection when

the IMF has a Y component. The magnetic field lines with azimuthally displaced footpoints map into a "banana" shaped convection cell in one hemisphere and an "orange" shaped cell in the other. Due to the Parker spiral our system is most often exposed to a $B_Y$ dominated IMF. The dipole tilt angle, varying between $\pm 34°$, leads to warping of the plasma sheet and oppositely directed $B_Y$ components in dawn and dusk in the closed magnetosphere. As a result of the Parker spiral and dipole tilt, geospace is most of the time asymmetric. The magnetic storm on August 17, 2001 offers a unique opportunity to study

the dynamics of the asymmetric geospace. IMF $B_Y$ was 20-30 nT and tilt angle was $23°$. Auroral imaging revealed conjugate features displaced by 3-4 hours magnetic local time. The latitudinal width of the dawnside aurora was quite different (up to $6°$) in the two hemispheres. The auroral observations together with convection patterns derived entirely from data indicate both dayside, lobe and tail reconnection in the north, but most likely only dayside and tail reconnection in the southern hemisphere. Increased tail reconnection during substorm expansion phase reduces the asymmetry.

*Copyright statement.*

## 1 Introduction

Over the last two decades it has been well established that the transverse component, $B_Y$, of the interplanetary magnetic field (IMF) leads to longitudinal displacement of the aurora in the conjugate hemispheres (Liou et al., 2001b; Østgaard et al., 2004, 2005; Wang et al., 2007; Liou and Newell, 2010; Østgaard et al., 2011b). As similar auroral features in the two hemispheres

can be considered as an illuminated footprint of conjugate magnetic field lines, these findings provide evidence of an"added"



$B_Y$ component in the closed magnetosphere with the same polarity as the IMF $B_Y$. This adds to the well known convection pattern asymmetry, with "orange" and "banana" cells due to IMF $B_Y$ (Heppner and Maynard, 1987), which are approximately mirrored in the two hemispheres. However, how a $B_Y$ component is established in the closed magnetosphere has been a controversy. It has often been referred to as a simple "penetration" of IMF $B_Y$ (Kozlovsky et al., 2003; Petrukovich, 2011;

Rong et al., 2015), while others have suggested that $B_Y$ is transported into the closed magnetosphere through tail reconnection (Stenbaek-Nielsen and Otto, 1997; Østgaard et al., 2004; Motoba et al., 2010). In this study we will show that increased tail reconnection has rather the opposite effect, it reduces asymmetry. In a recent paper Tenfjord et al. (2015) developed a more comprehensive understanding of how a $B_Y$ component is established in the closed magnetosphere. Adapting the ideas first suggested by Cowley (1981), but further developed to be consistent with observations by Khurana et al. (1996), they argued

that it is the asymmetric loading of magnetic flux in the lobes following dayside reconnection that induces a $B_Y$ component in the closed magnetosphere. It is the magnetic pressure force from the lobes that affects the closed magnetosphere and the term "penetration" of fields is misleading. Tenfjord et al. (2015) used magnetohydrodynamics (MHD) modeling to predict that the $B_Y$ component would be induced after just a few minutes. Tenfjord et al. (2015) also laid out a dynamical scenario of how the induced $B_Y$ component would lead to an asymmetric launch of Alfvénic waves and currents into the hemisphere where

the field line is connected to the "banana" cell. The ionospheric convection speed on the equatorward part of the "banana" cell was also predicted to be faster in order to restore symmetry closer to Earth. This was later confirmed by Reistad et al. (2016). In two follow-up papers Tenfjord et al. (2018, 2017), based on observational data from geosynchronous orbit (see also Wing et al. (1995)), showed that $B_Y$ was indeed induced on the closed field lines after only 5 to 10 minutes, similar to the response time between IMF orientation change and ionospheric convection change reported by Ridley et al. (1998) and Snekvik et al.

(2017). These results (Tenfjord et al., 2018) support the modeling results from Tenfjord et al. (2015) and contradicts the claims that reconnection is the mechanism by which a $B_Y$ component is transported into the closed magnetosphere. The latter would require about 50-60 minutes response time (Motoba et al., 2010; Rong et al., 2015) corresponding to the convection time from dayside to nightside across the polar cap. In the present study we will show that the role of increased tail reconnection is quite different. The importance of these studies are underpinned by the following: As the IMF orientation is distributed as a Parker

spiral, it can be shown that the strictly northward or southward dominated orientation, often studied, is a rather rare situation, while a $B_Y$ dominated IMF ($|B_Y| > |B_Z|$) is the common orientation ($> 70\%$ of the time). This means that most of the time we have asymmetric loading of magnetic flux, asymmetric footpoints of magnetic field lines, asymmetric aurora, currents and convection pattern. To put it briefly: geospace is most often asymmetric. To understand why and when, and how large the asymmetries are, is crucial for all studies involving mapping as well as any prediction efforts. During IMF $B_Y$ dominated

conditions both dayside, lobe and tail reconnection are expected to occur simultaneously (Reiff and Burch, 1985; Sandholt et al., 1998; Nishida et al., 1998), but the occurrence of lobe reconnection may also depend on the tilt angle (Crooker and Rich, 1993). The dipole tilt also leads to warping of the plasma sheet in the magnetotail (Tsyganenko, 1998, and references therein) and induces oppositely directed $B_Y$ components in the closed magnetosphere at dusk and dawn. As the dipole tilt angle is varying between $\pm 34°$ this is also a factor creating an asymmetric geospace.





The magnetic storm on August 17, 2001 offers a unique opportunity to study how all these effects are dynamically inter-related when we have a large IMF $B_Y$ component ($> 20$ nT), large tilt angle ($23°$) and two substorms with increased tail reconnection in their expansion phase. This magnetic storm was also studied by Longley et al. (2017b) who used conjugate auroral imaging to determine dawn-dusk offset of the polar cap between the hemispheres and compared those with four different

MHD model predictions. They found that none of the models reproduced the observed polar cap width. They also suggested that lobe reconnection was present in the northern hemisphere. In a follow up paper Longley et al. (2017a) also addressed the polar cap width, but focused more on the auroral data. They repeated the claim about seeing signatures of lobe reconnection in a DMSP pass in the northern hemisphere.

In the present paper we will use the same imaging data to identify discrete conjugate auroral features as well as sudden si-

multaneous brightening in order to study the asymmetric geospace during this magnetic storm. When using the term "discrete" we refer to distinct features produced by accelerated electrons that are sufficiently bright to be identified. They may be discrete arcs, but the spatial resolution of the cameras do not allow us to determine that unambiguously. This identification and the analysis we perform are based on the following assumptions:

a) As they are known to be associated with field-aligned currents, discrete auroral features produced by acceleration and

subsequent electron precipitation are a manifestation of coupling between the magnetosphere and ionosphere (see e.g., Ohtani et al., 2009). When identifying discrete auroral features in the conjugate hemispheres, we will look for similar shapes, not necessarily identical, but sufficiently bright to be identified. We emphasize that we do not intend to compare absolute auroral intensities, which could be affected by both different conductivity and different acceleration along field lines.

b) When auroral features with similar shapes and/or similar dynamical behavior can be identified simultaneously in both

hemispheres, it is a strong indication that these features have a common magnetospheric source region. As will be shown, we identify relatively bright poleward features indicative of tail reconnection, which by definition are magnetically connected. Although it is an open question how the electrons are accelerated to produce poleward boundary intensifications (Øieroset et al., 2002; Ohtani and Yoshikawa, 2016), there is observational evidence that bright auroral features at the nightside poleward boundary is found at the footpoint of reconnected field lines in the tail (Borg et al., 2007; Østgaard et al., 2009). We also

identify features that brighten up simultaneously in both hemispheres (a dynamical change, similar as substorm onset), which is also a strong indicator of having a common magnetospheric source and being magnetically connected.

With two hours of simultaneous conjugate auroral imaging we observed conjugate auroral features displaced by 3-4 hours MLT, similar as the 3 hours MLT reported by Reistad et al. (2016). The latitudinal width of the dawnside aurora was much larger in the northern than the southern hemisphere. Convection patterns derived from data only, combined with the auroral

features indicate dayside, lobe and tail reconnection in the northern hemisphere, but only dayside and tail reconnection in the southern hemisphere. As two substorms occurred during these two hours, we find that increased tail reconnection does not transport $B_Y$ component into the closed magnetosphere, but rather reduces the asymmetry by partly removing the magnetic pressure in the lobes. A similar reduction of asymmetry during substorm expansion phase was reported by Østgaard et al. (2011a).





After presenting our analysis of these asymmetric auroral features, we will discuss the implications of considering these features as *not* being conjugate, that they may result from either ionospheric processes, conductivity differences or have different sources in the magnetosphere.

## 2 Data

During the magnetic storm on August 17 2001 the IMAGE and Polar spacecraft provided 5 hours of imaging data from the same (northern) hemisphere and later more than 2 hours of simultaneous images from the conjugate hemispheres. The IMAGE Far UltraViolet instrument package provided images in three different wavelength bands (Mende et al., 2000), IMAGE-WIC: 140–180 nm, which includes the Lyman-Birge-Hopfield (LBH) N2 band and a few Nitrogen lines, IMAGE-SI12: Doppler shifted Lyman $\alpha$ (121.8 nm) and IMAGE-SI13: Oxygen emission (135.6 nm). The Polar VIS Earth camera (Frank et al., 1995) measured emissions in the 124–149 nm band, which is usually dominated by the atomic oxygen (O I) line at 130.4 nm but with contributions from oxygen line at 135.6 nm, the LBH N2 band and a Nitrogen line (Frank and Sigwarth, 2003), dependent on the electron energies (Frey et al., 2003a). To identify similar auroral features we have used IMAGE SI13 and Polar VIS and in Section 3 we discuss the importance of comparing images from cameras that are as identical as possible. Exposure times are 5 s, 10 s and 32.5 s and cadences are 123 s, 123 s and 54 s for IMAGE SI13, IMAGE WIC and Polar VIS Earth camera, respectively.

The solar wind and Interplanetary Magnetic Field (IMF) measurements are provided by ACE through the OMNI data (King and Papitashvili, 2005). They are time-shifted to the sub-solar bow shock location.

To establish convection pattern we use SuperDARN, SuperMAG and DMSP data. How the convection patterns are calculated will be explained in Section 3. CHAMP data are used to derive field aligned currents (Lühr et al., 1996).

There is a large amount of data that have been considered for the analysis presented in this paper. For transparency, we have uploaded a video showing the data coverage for the time interval from 15:50 UT to 18:59 UT. This comprises three channels of IMAGE data (SI12, SI13 and WIC), the Polar VIS images, SuperDARN line-of-sight coverage (green dots) and vectors from overlapping radars (green vectors), SuperMAG magnetic field perturbations rotated in the direction of an equivalent overhead current (brown vectors), DMSP and NOAA electron energy flux (pink bars), DMSP ion flow (green lines) and field-aligned currents derived from CHAMP data (red: upward current and blue: downward current). In this paper we will present a selection of the most prominent features in the imaging data as well as particle data and modeling results to support our interpretation.

## 3 Methodology

Simultaneous images from the conjugate hemispheres are used to identify asymmetric auroral features. The IMAGE FUV images were pre-processed using the FUVIEW3 software (http://sprg.ssl.berkeley.edu/image/) with the "corrected counts" option, ensuring that intensity across the detector (flat-field correction) and from different times of the mission (temperature correction) can be compared. For the SI13 camera we used an updated flat-field as the default flat-field did not per-



form well on the dayside during this event. The VIS Earth image data were downloaded from NASA's Space Physics Data Facility (ftp://cdaweb.gsfc.nasa.gov/pub/data/polar/vis/vis_earth-camera-full/) and processed using the XVIS 2.40 software (http://vis.physics.uiowa.edu/vis/software/), which includes a flat-field correction and the most updated values for pointing information.

The dayglow-induced emissions and background noise have been subtracted from each image separately. This is done by constructing a model of the dayglow emissions from pixels not influenced by aurora based on their solar zenith angle and satellite zenith angle in the mapped image. The modeled pixel intensity is then subtracted from all pixels leaving only auroral emissions in the image. The Poisson variation of the dayglow will still remain as noise in the image (Reistad et al., 2014).

    Ideally, we would use images from identical calibrated cameras for this, but unfortunately such auroral cameras in space do

not exist. The best we can do is to use cameras that at least detect emissions from the same constituents of the atmosphere. As VIS Earth and SI13 cameras are both detecting predominantly emissions from atomic Oxygen, we assume that intensity *changes* due to varying relative densities of Nitrogen versus Oxygen should be small. However, they are detecting different emission lines, namely 130.4 nm (VIS) and 135.6 nm (SI13), have different contributions from the LBH band and different scattering cross-sections, so we need to perform an inter-calibration as best we can. For this we use the 5 hours of data when

the two cameras were imaging the same aurora from the northern hemisphere and scale the intensities to display similar features. An example is shown in Figure 1. Although the VIS image has different pixel resolution than SI13 ($256 \times 256$ versus $128 \times 128$), the two images (Figure 1A and 1B) display similar auroral features with approximately the same intensities. The scaling between SI13 and VIS is not only based on this pair of images, but all the images when the two cameras were detecting the same aurora in the northern hemisphere (10 UT - 15 UT). The purpose is to establish a scaling that will be used for

comparing SI13 and VIS when they observed the aurora simultaneously in the northern and southern hemisphere, respectively.

    We also show the IMAGE WIC image, with two different color scalings, just to illustrate how differently the same auroral oval appears in a different wavelength band. If the two images from IMAGE-WIC and Polar VIS (Figure 1C and 1B), where the aurora in the dawn is scaled to match, had been from the conjugate hemispheres we would probably conclude that the dusk side aurora are fairly asymmetric in intensity. However, it is more likely that the dusk side aurora should be scaled to match

(Figure 1B and 1D) and that it is indeed the dawnside aurora that appears with much higher intensities in the WIC image than in the VIS image. Such a difference could then be explained by more energetic electrons drifting towards dawn scattered into the loss cone by waves and precipitate, which will appear more intense in the LBH band than in the 130.4 nm emissions (Frey et al., 2003a), as Figure 1D compared to 1B clearly shows. As WIC (LBH) and VIS (130.4 nm) respond differently to energetic precipitation, we will only use data from SI13 and VIS (with the scaling established in Figure 1) to identify conjugate auroral

features. There are also other technical issues that can lead to misinterpretation of auroral features and those are different viewing angles and dayglow subtraction. Auroral intensities obtained from very oblique viewing angles would be different than from nadir depending on the auroral structures, so care should be taken when looking at features that are imaged from slant angles. The dayglow removal can also introduce features or remove features. In our dayglow removal we make sure that what is left of counts equatorward of the dayside oval has a mean value of zero when averaged over a sufficiently large area.

Conductivity differences could also lead to misinterpretation as they are known to give different auroral intensities (Newell





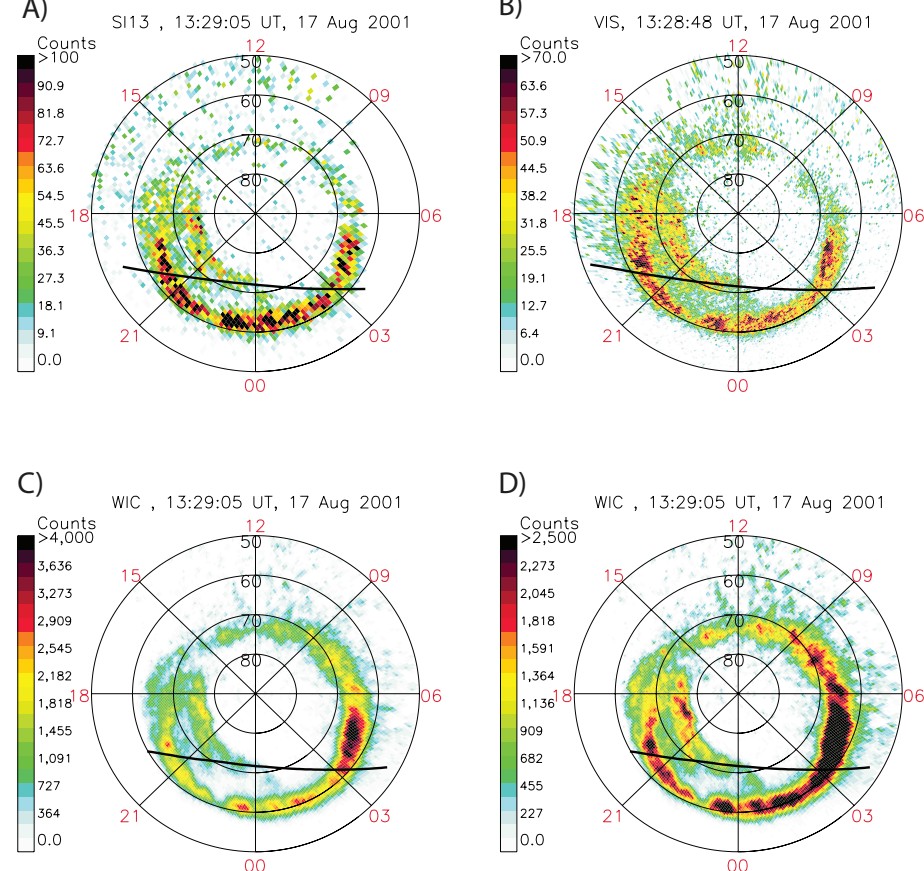

**Figure 1.** Images of different emission bands at 13:29 UT, August 17, 2001, separated ∼15 seconds (centre time). All images are from the northern hemisphere. Black lines indicate the terminator at 90° solar zenith angle. A) IMAGE SI13 B) Polar VIS C) IMAGE WIC where the dawn aurora is scaled to match VIS, D) IMAGE WIC, where the dusk aurora is scaled to match VIS.

et al., 1996; Liou et al., 2001a). However, we emphasize that we do not intend to compare absolute intensities, only discrete auroral features sufficiently bright to be identified.

To estimate the global plasma flow pattern (in a corotating frame), we adopt a novel purely data based multi-instrument approach. If the plasma is incompressible, i.e. if $\nabla \cdot \mathbf{v} = 0$, $\mathbf{v}$ can be expressed in terms of spherical elementary convection cells, similar to the much used spherical elementary current cell technique developed by Amm (1997). This was demonstrated in a local region, using SuperDARN measurements, by Amm et al. (2010). Here we use a global (poleward of 50°) grid of elementary convection cells, and estimate their amplitudes through a set of linear equations (see Amm et al., 2010, Section 2). As input we use three different datasets: 1) SuperDARN line of sight measurements of the plasma flow, providing one equation per measurements, 2) DMSP SSIES provides measurements of the vector flow, adding two equations per measurements (we use only measurements obtained within ±2 min of the time of the convection map), and 3) measurements from SuperMAG, of the





ground magnetic field, assuming that they correspond to currents opposite of the flow (Hall currents), and that 1 nT corresponds to 2 m/s. Laundal et al. (2015, 2016) showed that the assumption that the average equivalent current is the Hall current, and therefore can be used to find convection direction, is reasonable in sunlight (but only then). However, the conversion from 1 nT to 2 m/s is questionable, so we weigh the equations associated with these measurements by 0.3. That means that they only
become important in regions where no actual flow measurements are available. In addition to the measurements, we pad the circle at $50°$ with measurements of zero flow relative to a corotating frame, as a weakly imposed boundary condition.

The under-determined set of equations is solved by singular value decomposition, zeroing singular values that are smaller than $5\%$ of the largest value. We present the result in terms of the flow stream function $\psi$, which relates to the velocity by

$$\mathbf{v} = \hat{\mathbf{r}} \times \nabla \psi \tag{1}$$

where $\hat{\mathbf{r}}$ is a unit vector in the radial direction. Note that $\psi$ is not the same as the electric potential, yet they are similar.

## 4  The magnetic storm on August 17, 2001

We will now present the data obtained during the magnetic storm August 17, 2001.

### 4.1  A coronal mass ejection and solar wind conditions

On 15 August, 2001 at 23:54:05 UT GOES 8 measured a large increase in proton flux $> 100$ MeV (not shown) indicating a
Coronal Mass Ejection (CME). As is apparent from the time-shifted ACE data to the bow-shock a large pressure pulse was observed about 35 hours later: 17 Aug, 2001 at$\sim 11$ UT (Figure 2F) and immediately showed up in the SYM/H index (Figure 2H) as a huge increase of the magnetopause currents as a result of a dramatic compression of the magnetosphere. The arrival of this interplanetary shock and its effect on the magnetosphere at the initial phase of the storm has been reported by others (Huttunen et al., 2005; Echer et al., 2008).

The solar wind speed (Figure 2D) increased from 350 km/s to 500 km/s and stayed approximately constant for the next eight hours. The magnetic data revealed a large $B_Y$ component (Figure 2B) that stayed almost constant (20 nT–30 nT) and a varying $B_Z$ component (Figure 2C). For the time interval with conjugate imaging data available which we focus on in this study (16:45 UT to 19:00 UT) the IMF $B_Y$ was between 30 nT to 24 nT and $B_Z$ changed from 0 nT to -20 nT (clock angle from $90°$ to $135°$). This means that during this time interval (and also for hours before that) geospace was exposed to a strongly
$B_Y$ dominated IMF.

The solar wind conditions immediately caused intense magnetic disturbances, as can be seen from both AL index and SYM-H index (Figure 2G and 2H). The fluctuations seen around 11:00 UT in Figure 2D, 2E and 2F are not real, but an artifact of the OMNI data time shift to bow shock. During the two hours with simultaneous conjugate imaging there were two substorms, at 16:33 UT and 18:43 UT giving us an opportunity to study how increased tail reconnection during substorm expansion phase
affects the asymmetries.





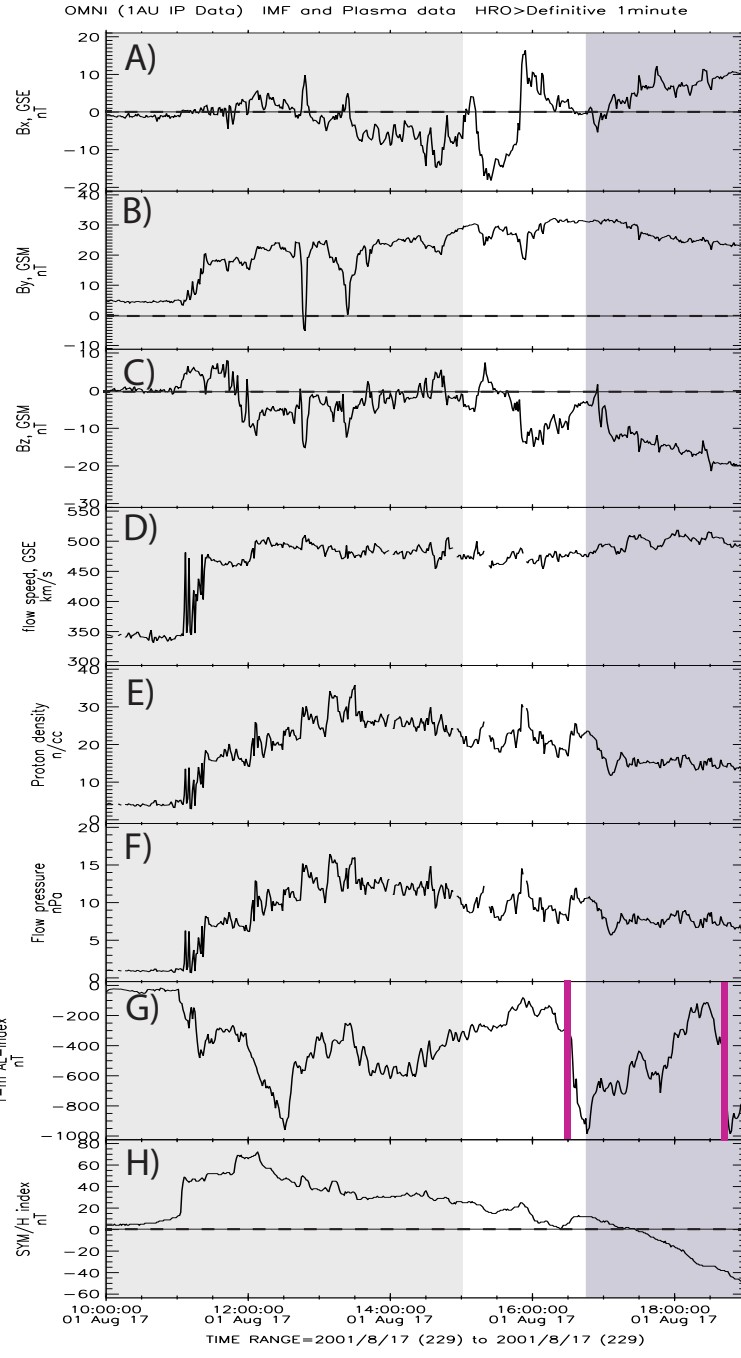

**Figure 2.** Solar wind (GSE coordinates), IMF (GSM coordinates) and geomagnetic indices during the passage of the CME on August 17, 2001. A) IMF $B_X$, B) IMF $B_Y$, C) IMF $B_Z$, D) solar wind bulk speed, E) proton density, F) flow pressure, G) AL index where the two vertical red lines are the times of two substorm onsets, and H) SYM-H index. The time period marked with light grey (10 UT to 15 UT) is when Polar and IMAGE were both viewing the same (northern) hemisphere, while the darker grey period from 16:45 UT to 19 UT is when the two satellites provided simultaneous images from the conjugate hemispheres.





## 4.2 Observations

Pairs of simultaneous images (SI13 and VIS, with the scaling established in Section 3) from the two hemispheres are shown in Figures 3 and 4. Features identified as conjugate are marked with red circles. Supporting particle data are shown in Figures 5 and 6.

In Figure 3A we identify a hook-shaped discrete auroral feature at the poleward edge in both hemisphere at the end of substorm expansion phase (Figure 2G) and interpret these as signatures of tail reconnection, and consequently being magnetically connected. They could be displaced by ∼1 hours MLT, but as the VIS FOV prevents us from seeing the duskward side of the aurora, it could be less. We also identify two north-south features, marked 1 and 2 (Figure 2A), seen in both hemispheres. We also believe these are conjugate and displaced by 0.5–1 hours MLT.

The next pair of images (Figure 3B) shows that the latitudinal width of the dawnside aurora is significantly larger in the southern hemisphere, where it extends to $80°$ magnetic latitude at 6 hours MLT. Since the auroral intensity in the SI13 is rather low, we show a DMSP 13 pass (Figure 5) through the dawnside aurora at 6.1 hours MLT in the north. The trajectory is shown by blue line in Figure 3B. From both the electron and proton spectrograms it is clear that the aurora does not extend beyond $74.1°$ magnetic latitude, which is a difference of $6°$ compared to the south. The difference in latitudinal width of the dawn aurora, although less pronounced, can also be seen in Figures 3C, 4A, 4B and 4C.

Now we return to Figure 3C, where we identify discrete auroral features at the poleward edge of the aurora (at the open-closed boundary) and we interpret these as signatures of tail reconnection. They are displaced ∼4 hours MLT and their poleward edges are about $4°$ different in magnetic latitude. We have also pointed to a weak auroral feature in the southern hemisphere at ∼22 hours MLT (red arrow) which could be interpreted as being conjugate to the eastward edge of the poleward feature in the northern hemisphere. One could then argue that it is a sensitivity issue that Polar VIS does not see the the entire poleward

arc from 18 MLT to 22 MLT. However, there are several arguments for this not being the case. First, the dynamical change that occurred from 18:11 UT (Figure 4A) to 18:13 UT (Figure 4B) when there is a sudden brightening at the eastward edge of the poleward feature in the north. In the southern hemisphere it is not the weak auroral feature marked with the red arrow in Figure 4B (VIS) that brightened up, but the region we identified in Figure 3C to be conjugate. When two relatively large regions brighten up simultaneously in the two hemisphere, it is a strong indication of having a common source region in the

magnetosphere. To further support our interpretation we show a NOAA 15 pass 4 minutes later at 18:17 UT (Figure 6) through the dusk-side aurora in the south. The trajectory is shown by blue line in Figure 4C. These electron data clearly indicate that there is no auroral precipitation poleward of $-62.2°$ magnetic latitude, while the northern aurora extends to about $70°$ magnetic latitude at 18.6 MLT, as seen in Figure 4C. It is not a camera sensitivity issue. There is no poleward auroral feature at 18-21

MLT in the southern hemisphere. In Section 4.7 we show that the data-derived convection pattern in the northern hemisphere, independent of the imaging data, locates the region of flux transport across the OCB (indicative of tail reconnection) exactly where the poleward auroral feature is bright. We will also show that the OCB is not moving equatorward during this interval to further support that there is indeed flux transport across the OCB in this region.



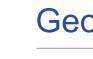 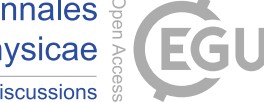

**Figure 3.** August 17, 2001: A-C) Pair of simultaneous (as close as possible) auroral images from northern (left) and southern (right) hemispheres. All panels show SI13 and VIS. Black lines indicate the terminator at 90° solar zenith angle. Features we claim are conjugate are marked with red circles and numbers in panel A. Blue line in panel B is the trajectory of the DMSP 13 pass in the northern hemisphere. Time of the first substorm is also shown.





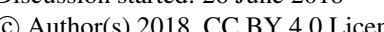

**Figure 4.** August 17, 2001: A-D) Pair of simultaneous (as close as possible) auroral images from northern (left) and southern (right) hemispheres. All panels show SI13 and VIS. Black lines indicate the terminator at 90° solar zenith angle. Features we claim are conjugate are marked with red circles. Blue line in panel C is the trajectory of the NOAA 15 pass in the southern hemisphere. Time of the second substorm is also shown.

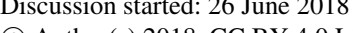
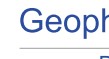
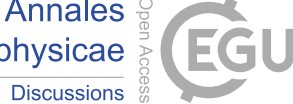


**Figure 5.** August 17, 2001: DMSP 13 pass through the dawn-side aurora at 17:20 UT in the northern hemisphere, see trajectory in Figure 3B.
A) Energy flux and average energies of electrons (black) and protons (red). B) Differential energy flux of electrons. C) Differential energy
flux of protons. Red vertical line marks the poleward edge of precipitation at 74.1° magnetic latitude.





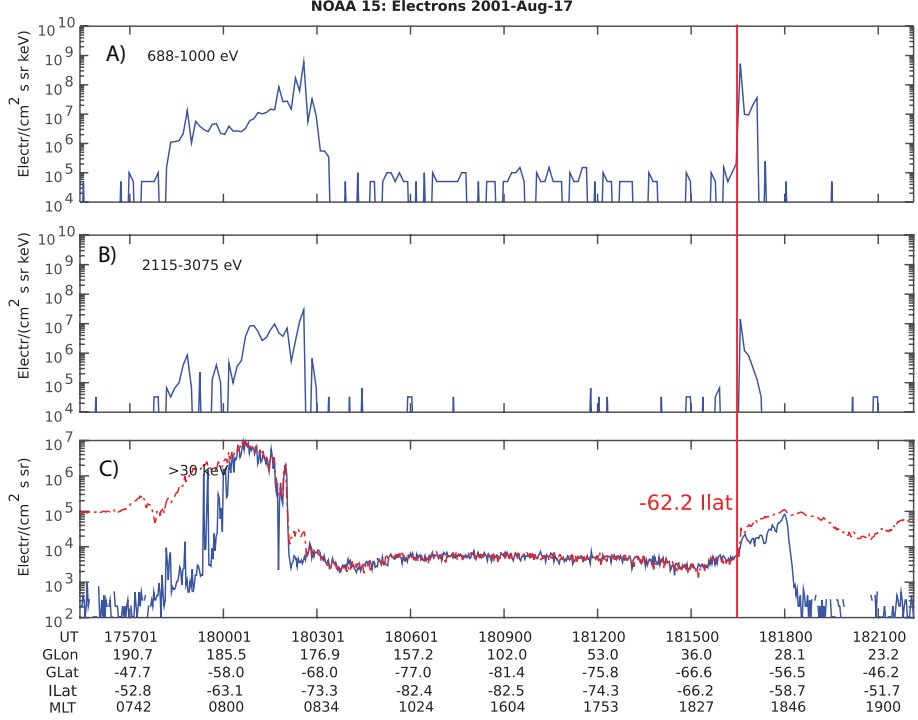

**Figure 6.** Particle measurements from NOAA 15 pass in the southern hemisphere on August 17, 2001 , see trajectory in Figure 4C. It passed through the dusk-side aurora at 18:17 UT. A) Electron flux: 688-1000 eV. B) Electron flux: 2115-3075 eV. C) Electron flux: $\geq 30$ keV, where blue line shows electrons within the loss cone and red line shows the locally mirroring electrons. Red vertical line shows the poleward edge of precipitation at $-62.2°$ magnetic latitude.

Our last pair of images (Figure 4D), ten minutes after the second substorm, shows relatively large north-south discrete auroral features that are not identical but have similar shapes. These features are in darkness in both hemispheres, so conductivity differences should not be an issue. Based on how similar they are, and that there are no other features that could confuse the link we make between these features, we interpret these also to have a common magnetospheric source region and consequently

5  they are magnetically connected. The displacement is about 1-1.5 hours MLT.

### 4.3  Comparison with model

As was seen in Figure 3C the conjugate auroras in the two hemispheres are displaced by about 4 hours MLT. From careful inspection of auroral features before and after we are fairly confident that this interpretation is correct. In order to compare our observations with an empirical model, we have used the Tsyganenko 02 model to investigate the magnetic mapping between

10  the hemispheres (Tsyganenko, 2002a, b). In addition to the tilt angle of $23°$ the input parameters (as seen from Figure 2) should have been: dynamic pressure: 8 nPa, solar wind speed: 500 km/s, Dst: -17 nT, IMF $B_Y$: 26 nT and IMF $B_Z$:-14 nT. As these IMF conditions are rather extreme, especially the dynamic pressure, $B_Y$ and $B_Z$, and in a range very poorly represented in the



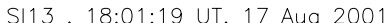

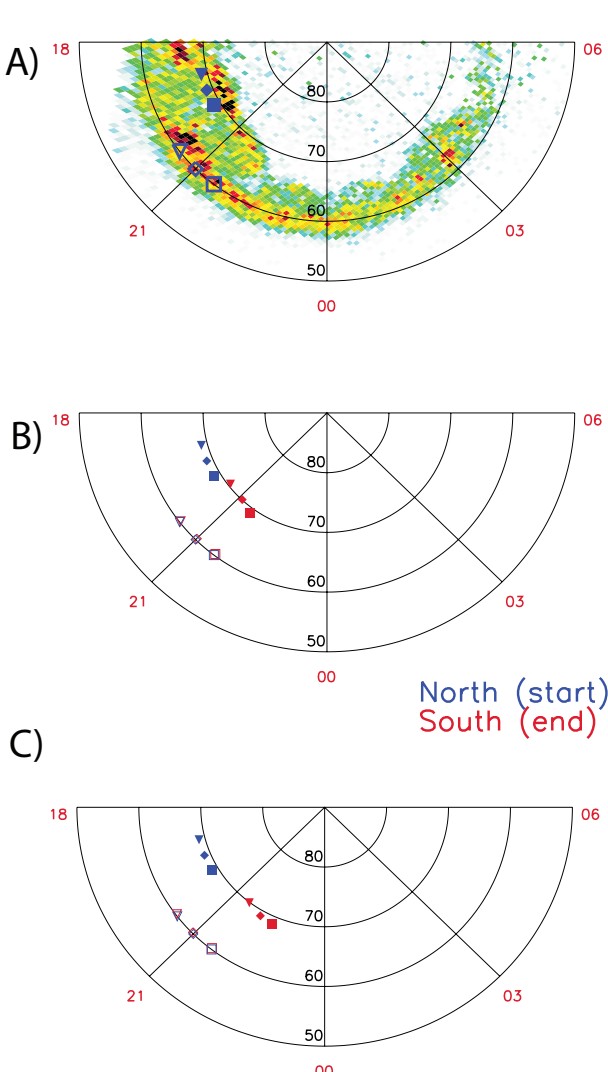

**Figure 7.** Field line tracing from one hemisphere to the other using the Tsyganenko 2002 model. A) Blue symbols indicate the starting tracing points in the northern hemisphere. Filled and open symbols are used for the poleward and the equatorward edge of the aurora, respectively. B and C) Blue (red) symbols mark the footprint in the northern (southern) hemisphere for tracing from north to south using different input values (see text for explanation).





database used to generate the T02 model, the output would be an extreme extrapolation of the model. We have therefore reduced the input values to: dynamic pressure: 2 nPa, IMF $B_Y$: 10 nT and IMF $B_Z$:-5 nT (clock angle is unchanged) but kept the others as measured (Figure 7B). In addition we show the results for input values of dynamic pressure of 2 nPa, but 50% larger IMF $B_Y$: 15 nT and IMF $B_Z$:-8 nT (Figure 7C) but still with same clock angle. The latter input values are used to show how the

extrapolation of the model leads to even larger asymmetries and serves to illustrate the combined effect of large IMF $B_Y$ and large tilt angle. The starting points in the northern hemisphere are chosen as far north as possible (69° magnetic latitude) in order to return closed field lines within the region where the model is valid. We have chosen the point on the westward side of the poleward arc, see filled blue symbols in Figure 7A. The starting points at the equatorward edge are marked with open blue symbols. The red symbols in Figure 7B and 7B are the mapping locations in the southern hemisphere. The tracing shows a

large asymmetry on the poleward edge of the aurora, about 1.5 hours MLT in Figure 7B and 2.5 hours MLT in Figure 7C and close to no asymmetry at the equatorward edge. This increase of asymmetries for increasing IMF $B_Y$, but same clock angle, is expected from our understanding of how asymmetries are produced by asymmetric loading of magnetic flux in the lobes, and how asymmetries are reduced as the flux tubes move toward the Earth (Tenfjord et al., 2015). We want to emphasize that we do not expect the T02 model to reproduce exactly our findings, but it gives qualitatively support for observing large asymmetries

at the poleward edge of the auroral oval at dusk.

### 4.4   Why so large asymmetries?

As suggested by Hau and Erickson (1995) and Khurana et al. (1996) a $B_Y$ component will be induced in the plasma sheet (closed magnetosphere) as a result of asymmetric loading of magnetic flux to the lobes and subsequent plasma flow (or motion of magnetic flux tubes), as shown in Figure 8A. This effect is apparent in the Tsyganenko model (Østgaard et al., 2005) and

MHD models (Tenfjord et al., 2015). As shown in Figure 8B there will be an additional effect of a large tilt angle. For positive tilt angle the central plasma sheet will move upward but less so towards the flanks. This configuration has been described as warping of the plasma sheet (e.g. Gosling et al., 1986, and references therein). Similar to the asymmetric loading of magnetic flux in the lobes leading to plasma flow and induced $B_Y$, the warping of the tail also causes asymmetric flow of magnetic flux towards the plasma sheet and induces $B_Y$ components in the closed magnetosphere of opposite polarity in dawn and dusk

(Tsyganenko, 1998; Liou and Newell, 2010, and references therein).

We interpret the large displacement of the footpoints for conjugate field lines that we observe (3-4 hours MLT) in the dusk sector to be the combined effect of large IMF $B_Y$ and large tilt angle/warping. In the dawn sector the warping effect will reduce the effect of IMF induced $B_Y$ and this is consistent with what we see in Figure 3A for feature 1 and 2.

### 4.5   Why wider aurora in the southern dawn?

As we pointed out in Section 4.2 the auroral oval is about 6° wider in the southern dawn compared to northern dawn (Figure 3B). We interpret this also to be an effect of the asymmetric loading of flux (due to IMF $B_Y$) and maybe an additional effect of enhanced lobe pressure in the north from warping for large tilt (Figure 8B). The effect is only on the poleward side of the aurora, while the equatorward boundary of the aurora is at similar latitudes in the two hemispheres. If we assume that the dawn aurora




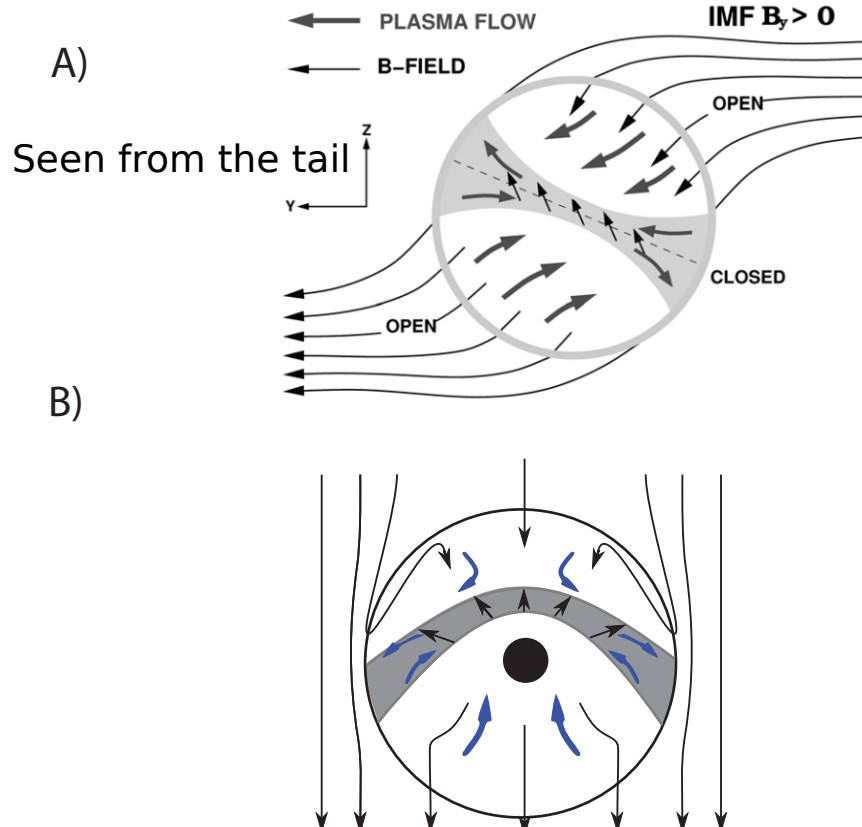

**Figure 8.** Plasma flow and magnetic fields in the mid tail (YZ plane) as seen from the tail. The white regions are the lobes and the shaded grey regions are the plasma sheet in the closed magnetosphere. A) The effect of IMF $B_Y > 0$, similar to Figure 3a in Liou and Newell (2010) which was adapted from Khurana et al. (1996). B) The effect of large tilt angle and warping of the tail (Tsyganenko, 1998), similar to Figure 3b in Liou and Newell (2010). The plasma flow (or the motion of magnetic flux tubes) are shown as thick grey and blue arrows in A and B, respectively. Magnetic field are shown as thin black arrows.




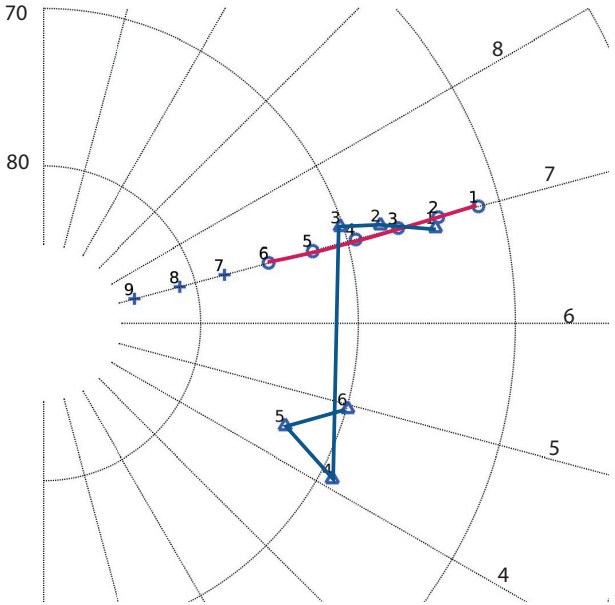

**Figure 9.** Tracing field lines from south (red), $62°$ to $75°$ at 7 MLT, to north (blue) using the LFM model. The field lines from $70°$ to $75°$ latitude in the south map to an azimuthally extended region in the north.

is on conjugate field lines the magnetic flux in the auroral oval has to be conserved. This means that the poleward segment of the aurora in the south has to map to an azimuthally extended region in the north. In order to justify such an interpretation we ran the Lyon-Fedder-Mobarry (LFM) MHD model (Lyon et al., 2005; Merkin and Lyon, 2010) with boundary conditions similar to the August 17 event. From Figure 9 we see that field lines from $70°$ to $75°$ latitude (at same longitude) map to an

azimuthally extended region in the north. At least qualitatively, the LFM model supports this interpretation.

**4.6    Time evolution of asymmetries and the role of substorms**

In Figure 10 we display the evolution of asymmetric locations of conjugate auroral features in the two hemispheres. For the first point (at 16:48 UT) we show both the feature within the red circle in Figure 3A ($\Delta MLT \sim 1$, with an approximate uncertainty as explained in Section 4.2) as well as the equatorward tip of features 1 and 2 ($\Delta MLT = 0.5$). We see that the asymmetries

are smaller after each substorm and build up to 3-4 hours MLT between the substorms. We interpret this as follows. It is well established by observations that the lobe pressure increases before substorm and decreases in less than one hour after substorm onset (see e.g. Caan et al., 1975). The accumulation of open flux before substorm causes the magnetosphere to inflate and the tail magnetopause flares outward. This increases the cross-sectional area that the magnetosphere presents to the solar wind and leads to build-up of pressure in the tail. When this loading is asymmetric, due to the large IMF $B_Y$, the induced $B_Y$

increases before substorm. As has been shown statistically by Juusola et al. (2011) the rate of fast bursty bulk flows increases significantly during the substorm expansion phase (see their Figure 4). As bursty bulk flows are signatures of tail reconnection,





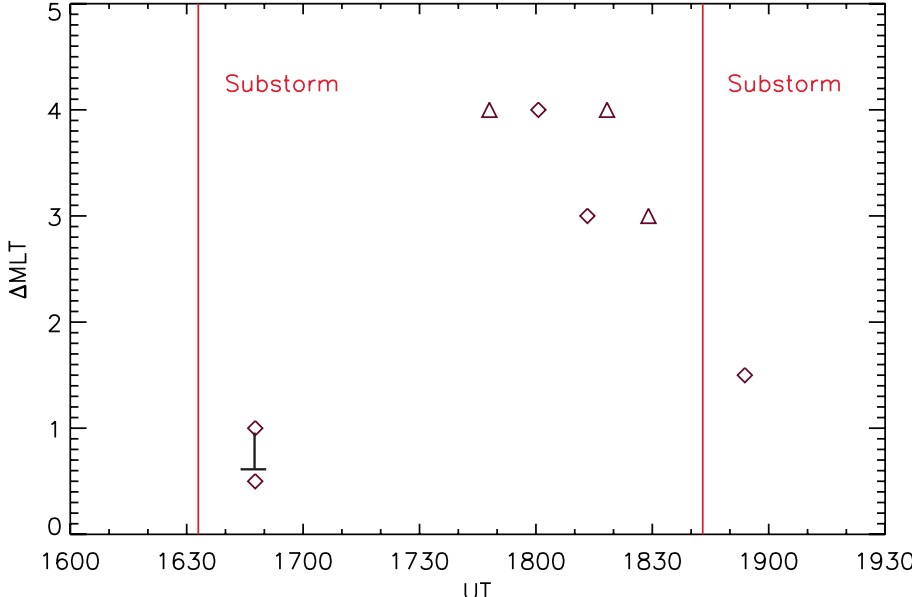

**Figure 10.** Time evolution of asymmetric conjugate auroral features. Diamonds are auroral asymmetries identified from Figure 3 and 4, while triangles are identified from images not shown, but can be found in auxiliary material. The red vertical lines are the times of the substorms.

it means that open magnetic flux is efficiently closed in the tail and the flaring angle decreases. This results in a smaller cross-sectional area of the magnetosphere leading to a lower pressure in the tail. The asymmetric lobe pressure that induced the asymmetries in the first place is effectively reduced when tail reconnection increases. Contrary to what have been suggested by others (Stenbaek-Nielsen and Otto, 1997; Østgaard et al., 2004; Motoba et al., 2010), increased tail reconnection does not

add any $B_Y$ component into the closed magnetosphere, but by reducing the asymmetric lobe pressure rather acts to reduce the $B_Y$. Observed reduction of asymmetries during substorm expansion phase was also reported by Østgaard et al. (2011a).

### 4.7 Convection pattern

Finally, as outlined in Section 3, we will use all available data to find the most likely convection patterns as well as the location of the open-closed boundaries in the two hemispheres for the period we observe large asymmetry. The convection patterns are

derived independently of the imaging data. Although 18:01 UT is the time when the asymmetries are most clearly identified, we have chosen the time interval around 17:47 UT for this purpose, because we have data from two SuperDARN radars with overlapping FOV to help us draw the convection pattern in the southern hemisphere. The bright poleward auroras indicative of tail reconnection are still clearly seen in both hemispheres. In the northern hemisphere we also have good data coverage at this time and can use both SuperDARN line-of-sight, a DMSP SSIES and SuperMAG data as input to the spherical elementary

convection cells technique described in Section 3. We also want to point out that we have good data coverage in the northern



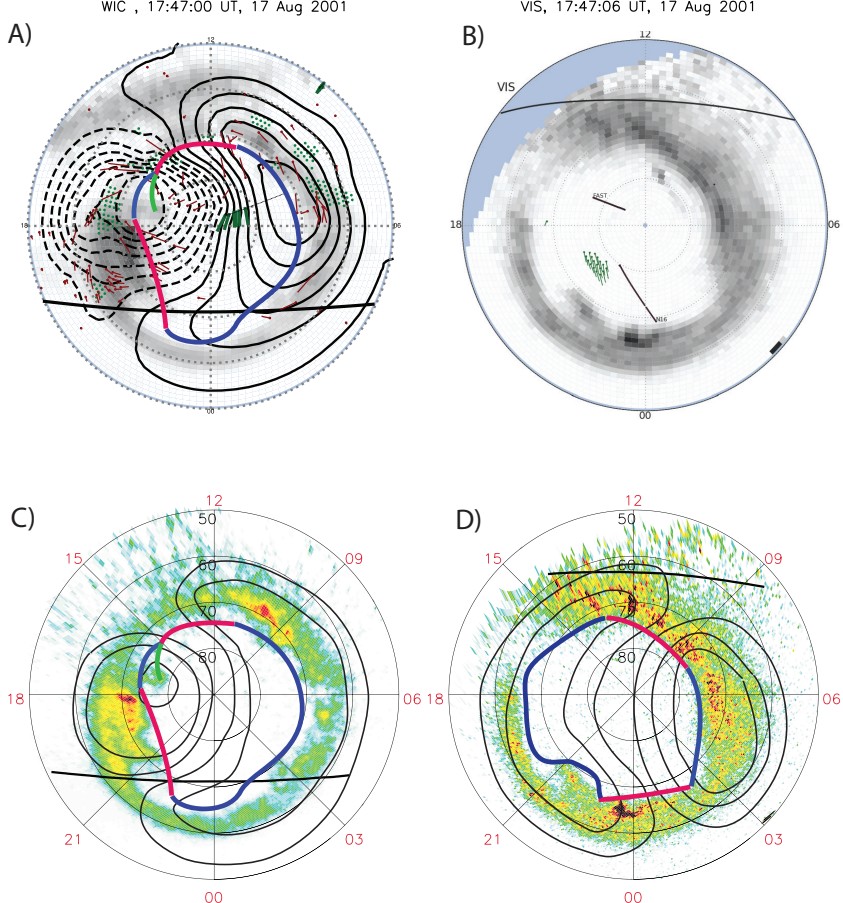

**Figure 11.** Convection patterns in the two hemispheres at 1747 UT, August 17, 2001. A) SuperDARN line-of-sight (green dots), SuperMAG (brown arrows) and DMSP SSIES (green arrows) are used to estimate the global convection pattern in the northern hemisphere. Blue and red lines indicate open-closed boundary, where the red lines are dayside and nightside reconnection regions. Green line indicates lobe reconnection. B) Southern hemisphere where a few vectors from SuperDARN are available (green arrows) C) Similar convection pattern as from panel A (now drawn by hand) overlaid the auroral intensities. D) Convection pattern and open-closed boundary in the southern pattern drawn by hand, guided by the few available SuperDARN vectors (panel B) and auroral features of dayside and nightside reconnection.

hemisphere both before and after 17:47 UT and the derived convection pattern does not change, when using data input before and after that time.

### 4.7.1 Convection pattern in the northern hemisphere.

In Figure 11A we show the data-derived convection patterns in the northern hemisphere and it reveals a relatively small "orange" cell in the dusk and a large "banana" cell in the dawn. We use the poleward boundary of the aurora as a proxy for the open-closed boundary (OCB) (Laundal et al., 2010), which is marked as blue and red lines. The red lines indicate where





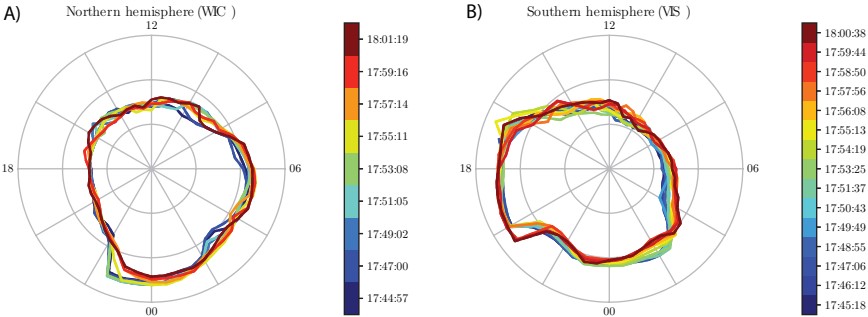

**Figure 12.** Open-closed boundaries derived from IMAGE WIC images in the north and Polar VIS in the south from 17:45 UT to 18:01 UT on August 17, 2001.

plasma, according to the derived convection pattern, flows across the OCB. We can identify dayside reconnection between 11 MLT and 15 MLT that opens magnetic flux and tail reconnection between 18 MLT and 22 MLT that closes magnetic flux. In Figure 12 we show the time evolution of the the OCB in the two hemisphere derived quantitatively from IMAGE WIC (450 counts) in the north and Polar VIS (5 counts) in the south. From Figure 12A it can be seen that the OCB in the northern

hemisphere does not move with the flow in the 11-15 MLT and 18-22 MLT sectors from 17:45 UT to 18:01 UT, but is rather stable, which means that there is indeed flux transport across the parts of the OCB marked as red lines. For the region of tail reconnection (18-22 MLT in Figure 11A), this is exactly where we observe the bright poleward feature in the north, giving independent support for our interpretation that the poleward auroral feature is indeed a signature of tail reconnection (Section 4.2).

We have marked lobe reconnection by a green line from 15-17.5 MLT. The observational support for lobe reconnection is the auroral spot seen at 17.5 MLT in Figure 11A and 11C between $75°$ to $80°$ magnetic latitude, clearly detached from the oval. This feature $> 75°$ magnetic latitude is present in almost all the images since the arrival of the CME at 11 UT. It is seen during the 5 hours of imaging data from the same hemisphere and from 16:18 UT when we have images from conjugate hemispheres (Figure 3A). The spot becomes faint at 17:15 UT but reappears between 17:40 UT and 17:50 UT (Figure 11C). Figure 13 shows

IMAGE WIC images (northern hemisphere) from 16:19 UT to 16:49 UT, where this spot is seen clearly detached from the auroral oval in all the ten images (see also the video, uploaded as auxiliary data). Frey et al. (2003b, 2004) explored statistically the characteristics of this high-latitude dayside aurora (HiLDA) and concluded that the major driving process was high-latitude (lobe) reconnection and that it occurred preferentially in the summer hemisphere.

To further support that the auroral spot is associated with lobe reconnection we show the images as well as the derived

convection pattern from the northern hemisphere at 16:31 UT (Figure 14). The spot is seen in the SI13 and WIC but not in SI12, which means that it is produced by electron precipitation. This is further supported by the derived upward current from CHAMP data. The derived convection shown by blue arrows in Figure 14 indicates sunward flow at very high latitudes ($75°$), just where we see the spot and the upward field-aligned current from CHAMP. The LFM model also predicts a strong upward current on open field lines between $70°$ and $80°$ at 17-18 MLT (not shown).





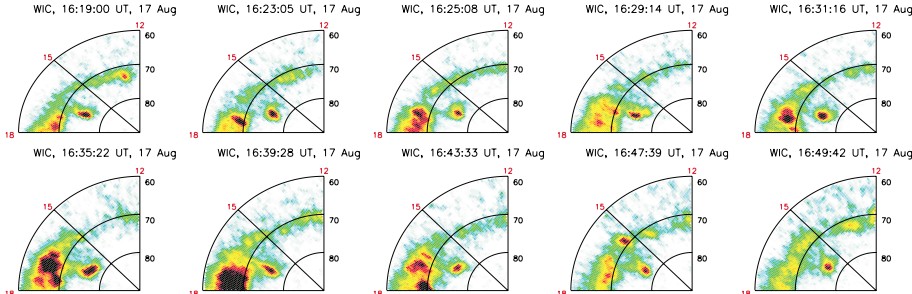

**Figure 13.** IMAGE WIC images. The auroral spot $> 75°$ magnetic latitude, clearly detached from the auroral oval and indicative of lobe reconnection, is seen from 16:19 UT to 16:49 UT, August 17, 2001.

Flow across the OCB (Figure 14) indicating dayside reconnection is seen around 15 MLT and $< 70°$ with downward current (CHAMP) and proton precipitation (SI12 image).

Flow across the OCB is also seen at 11 MLT and $< 70°$ consistent with the dayside reconnection region we have indicated by the red line in Figure 11A. The equatorward convection just outside the auroral oval near noon is probably an artifact of the

inversion, since the solution there is relatively unconstrained by observations compared to surrounding regions (SuperDARN backscatter locations are indicated by green dots). To have lobe reconnection under such IMF $B_Y$ dominated conditions has been reported by others (Sandholt et al., 1998; Nishida et al., 1998) and especially in the northern hemisphere for such large tilt angle (Crooker and Rich, 1993). The combination of dayside reconnection and lobe reconnection was suggested by Reiff and Burch (1985) for weakly southward IMF with dominated $B_Y$, which is exactly the IMF conditions we have. Compared to

their Figure 1, our red line corresponds to their region L to M, and our green line (lobe reconnection) to their region between M and H. The statistical studies by Frey et al. (2003b, 2004) also support our interpretation. In Figure 11C we have drawn (by hand) the same convection pattern and open-closed boundary onto the WIC color image from 17:47:00 UT.

### 4.7.2 Convection pattern in the southern hemisphere.

Now we return to Figure 11B and 11C to suggest a possible convection pattern in the southern hemisphere. In this hemisphere

there are too few data points to apply the spherical elementary convection cells technique. However, we can use the auroral features (Figure 11D) combined with the convection vectors (green arrows in Figure 11B) measured by two SuperDARN radars with overlapping FOV to determine the OCB and suggest a possible convection pattern. Firstly, we notice the auroral signature of dayside reconnection between 9 MLT and 12 MLT. This is even more clear in the VIS image from 18:01 UT (Figure 3C). Secondly, as we argued in Section 4.2, the discrete poleward feature in the southern hemisphere between 22 MLT and 2 MLT is

the auroral signature of tail reconnection in this hemisphere. As for the northern hemisphere we mark the OCB at the poleward side of the auroral oval (see also Figure 12B) with red line indicating reconnection regions and blue line for the rest of the OCB. With these constraints and SuperDARN vectors to guide us we have drawn one possible solution for the convection pattern overlaid the VIS image from 17:47 UT (Figure 11D). In the southern hemisphere we do not see any features that could





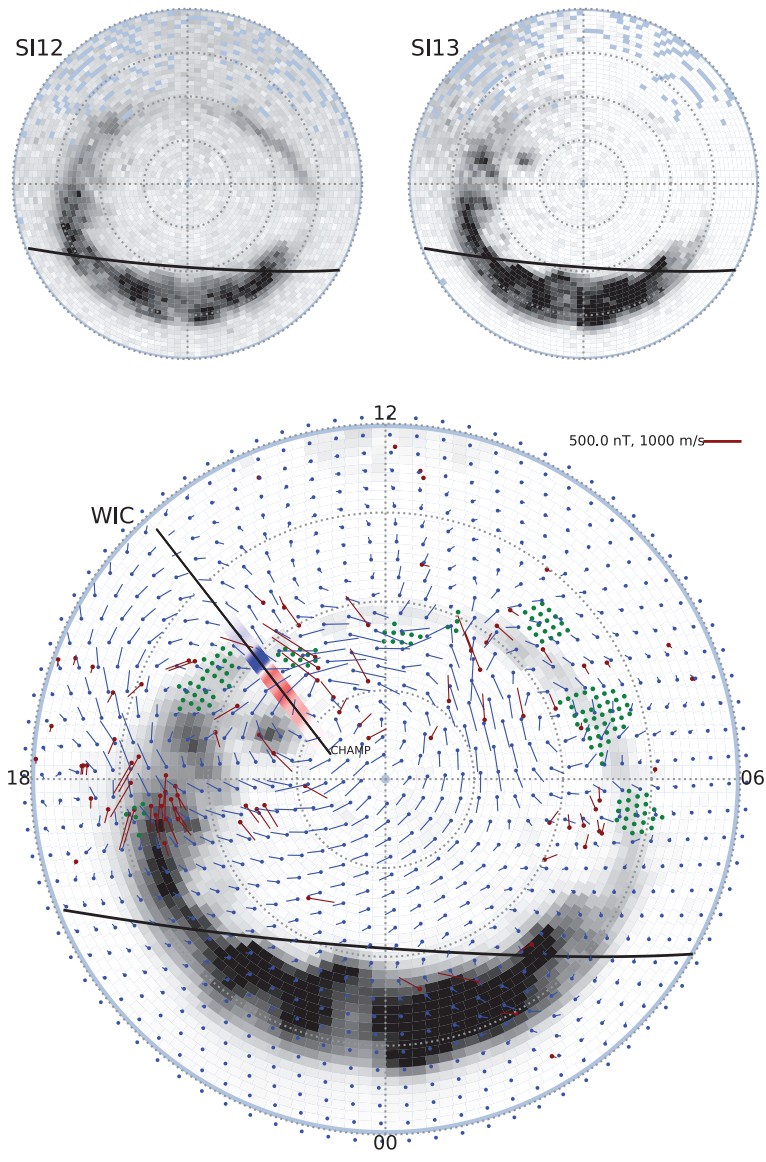

**Figure 14.** Auroral images from the northern hemisphere at 16:31 UT, August 17, 2001. A) SI12, B) SI13 and C) WIC with derived currents from CHAMP pass. Red (blue) is upward (downward) currents. The green dots indicate locations of SuperDARN backscatter. SuperMAG equivalent currents are shown with brown vectors. The scale of the convection and magnetic field perturbations are shown in the top right corner of the WIC image. The location of the sunlight terminator is indicated by a black line. Derived convection pattern are shown by blue vectors.




indicate lobe reconnection. We believe that this is due to the very large tilt angle (23°) which makes the northern lobe more exposed to the IMF than the southern lobe, and can explain why we see lobe reconnection in one hemisphere and not in the other. This is in agreement with Crooker and Rich (1993) who suggested that lobe reconnection is a summer phenomenon that can occur in one hemisphere only and Frey et al. (2003b, 2004) who found the HiLDA also to be a summer phenomenon.

## 4.8 Alternative interpretation

Let us now consider that the features we identify as being asymmetric are *not* conjugate, but result from either ionospheric processes or have different sources in the magnetosphere: We have identified three types of features:

1. Intensification of the aurora close to the open-closed boundary, often called poleward boundary intensification (PBI), Figure 3A and 3C.

2. Large scale sudden brightening (Figure 4B).

3. Large scale bright auroral structures with similar shapes (Figure 4D).

PBIs are often being associated with tail reconnection. However, in a recent paper Ohtani and Yoshikawa (2016) propose a mechanism where the PBIs result from convection flow shears in the ionosphere. The idea is that when "fast polar cap flows", which is the dynamical driver in this scenario, encounters the poleward boundary of the aurora the conductivity gradients associated with this aurora forces the flux tubes to move along rather than across the boundary. This results in a flow shear which creates a pair of upward/downward currents and consequently auroral intensification at the upward leg. However, as these "fast polar cap flows" are open flux tubes dragged across the polar cap by the solar wind, a fast flow approaching the OCB also means an increase of inflow in the tail reconnection region. Such a flow can only be sustained if field lines reconnect in the magnetosphere. We do not question that the fast flows leads to flow shears in the ionosphere and that this may explain the changes in auroral intensities, but we will argue that the ultimate driver for these "fast polar cap flows" is reconnection at the tailward end of these flux tubes. These "fast polar cap flows" and associated auroral intensifications are ionospheric signatures of enhanced reconnection and consequently conjugate.

Let us now assume that the poleward auroras in Figure 3C have two different magnetospheric sources and the magnetosphere is less asymmetric or even symmetric. We will focus on the 18-22 MLT region. In this region we would then need a mechanism that launches particles (preferentially electrons) close to the OCB only into the northern hemisphere. Figure 6 clearly shows that there is a complete absence of particle precipitation in the southern hemisphere at 18.6 MLT at high latitudes. We are not aware of any theory that can explain a relatively high flux of particles into one hemisphere and total lack of particles into the other hemisphere from the same large source region (about 4 MLT hours) in the magnetosphere.

To further examine whether the 18-22 MLT sector in the two hemispheres might be conjugate we show (Figure 15) the average counts/pixel between 55° and 75° magnetic latitude in this sector. No scaling between cameras is applied as we only want to explore the intensity changes in each sector. Northern hemisphere is shown by the blue line and southern hemisphere by the orange line. The linear Pearson correlation coefficient is only 0.002. We also show the average counts/pixel in the 23-3 MLT sector in the southern hemisphere (green line). The correlation coefficient between the blue and green line (that is the





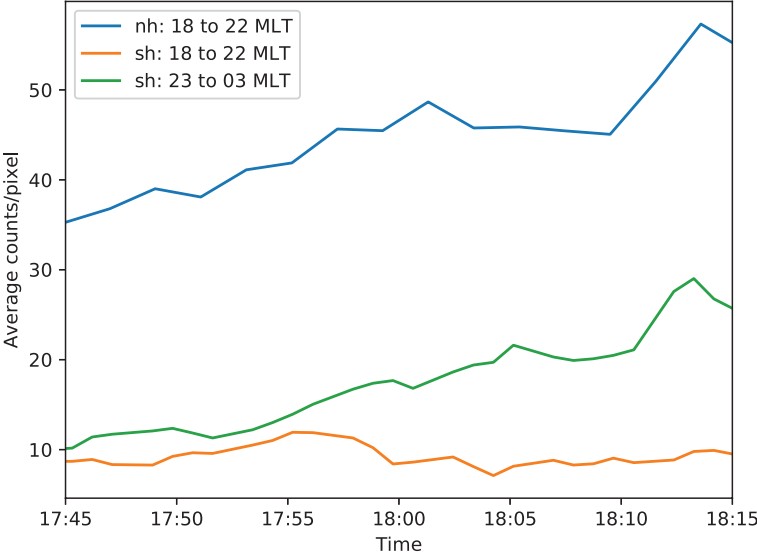

**Figure 15.** Average counts/pixel between $55°$ and $75°$ magnetic latitude in selected MLT sector from 17:45 UT to 18:15 UT on August 17, 2001. Blue line: Northern hemisphere 18-22 MLT, Orange line: Southern hemisphere 18-22 MLT and Green line: Southern hemisphere 23-3 MLT.

two sectors we claim to be conjugate) is 0.95. These sectors show a steady increase and a spike at 18:13 UT, which are the sudden brightening (seen in Figure 4B).

An often cited mechanism for differences in auroral intensities is the suppression of discrete aurora by sunlight (Newell et al., 1996). The poleward aurora seen in Figure 3C in the 18-22 MLT sector in the north is indeed in sunlight, but much brighter than the (missing) aurora in the southern darkness between 18-22 MLT. This is opposite of what Newell et al. (1996) predict.

We also want to point to the convection pattern (Figure 11 and 14) that independently of imaging data shows strong convection across the OCB (reconnection) exactly where we see the poleward aurora in the north. If we assume that the northern 18-22 MLT region maps to the 18-22 MLT region in the south (symmetric magnetosphere) we would expect to see at least some "weak" signatures of this tail reconnection in the south. We therefore find it quite unlikely that the tail reconnection region maps to 18-22 MLT in the southern hemisphere, but find strong support for this to be the case in the northern hemisphere. We conclude that both particle data (Figure 6) and convection pattern (Figure 11 and 14) contradict the hypothesis of a symmetric magnetosphere. What could then be questioned is whether the tail reconnection maps to the 23-3 MLT region in the south. Unfortunately, we do not have convection data showing flow across the OCB in this region, but it is hard to envision any other auroral features that could indicate tail reconnection in the south.

The sudden brightening seen in Figure 4B is about 3 hours MLT displaced in the two hemispheres. The aurora in both hemispheres brightens up in less than 2 minutes and is accompanied by a 100 nT decrease in the AL index. Although the brightening does not expand into typical substorm bulges they both have extension of $6 - 8°$ latitude and more than 1 MLT





in azimuth. Mapped into the plasma sheet they cover huge regions. If these are two separate regions, we have one large magnetospheric source region that only launches electrons into the northern hemisphere (and nothing to the southern hemisphere) and another large magnetospheric region that, at the same time, only launches electrons into the southern hemisphere (and nothing into the northern hemisphere). Again, we are not aware of any theory that could explain such asymmetric behavior

of the magnetosphere. Alternatively, there is a large region in the magnetosphere that maps to a 5 hours MLT sector in the ionosphere (21-2 MLT) and that the western part launches electrons into the northern hemisphere and the eastern part only launches electrons into the southern hemisphere. The same scenario could then be used for all substorm onset studies and we are not aware of any studies that have found support for such an interpretation. Although not very likely, we cannot rule out this possibility.

Finally, if we assume the auroral features in Figure 4D to have different magnetospheric source regions, we end up with the same dilemma as for the sudden brightening. One source region only launch electrons into one hemisphere and another only into the other hemisphere.

## 5   Conclusions

The magnetic storm on 17 august 2001 has given us a unique opportunity to study the dynamical behavior of the asymmetric

geospace. The magnetic storm occurred at 11 UT on August 17, 2001. Solar wind had high pressure and IMF was dominated by a large $B_Y$ of 20-30 nT and clock angle varied between $\sim 90°$ and $\sim 135°$. From 16 UT to 18 UT the tilt angle was as large as $\sim 23°$. Based on conjugate imaging and all other available data during this event, we find that the combination of a large IMF $B_Y$ component and large tilt angle sets up a highly asymmetric system. Furthermore, having two substorms during this period, we are able to address how increased tail reconnection during substorm expansion phase affect the system.

Considering all the available data we have presented a plausible interpretation and our main findings can be summarize as follows:

1. The auroral images show extremely asymmetric conjugate footpoints implying a very asymmetric magnetosphere. The asymmetries are supported qualitatively by the Tsyganenko 2002 model.

2. Dusk asymmetries up to 3-4 hours MLT (at the poleward edge) can be explained by the combination of two effects: large

IMF $B_Y$ (20-30 nT) and large tilt angle (23°). The latter leads to warping of the plasma sheet. Combined, they induce a large $B_Y$ in the closed magnetosphere at dusk.

3. Dawn asymmetries are smaller because large tilt/warping reduces the effect of IMF $B_Y$. The net induced $B_Y$ on closed field lines is smaller in dawn than dusk.

4. The wide oval in the southern dawn and the narrow oval in the northern dawn are also consistent with asymmetric loading

of magnetic flux that increases the lobe pressure. MHD modeling results qualitatively support that the poleward region of the wide oval maps to an azimuthal extended region in the north, consistent with conservation of magnetic flux.



5. Convection patterns and a persistent high latitude auroral spot (HiLDA) indicate that signatures of dayside and tail reconnection were seen in both hemispheres, while lobe reconnection only occurred in the northern hemisphere that is tilted towards the solar wind; the summer hemisphere.

We have also discussed other interpretations, that the asymmetries are due to conductivity differences, ionospheric processes or have different sources in the magnetosphere. We find that they are either not supported by data, or they require some unknown processes of launching electrons into only one hemisphere.

Although this is a rather extreme event, it serves as a good illustration of how important it is to consider geospace as an asymmetric system. Ignoring this could lead to large errors and misinterpretations in MI-coupling studies. However, it should be noticed that even for smaller values of IMF $B_Y$ large asymmetries have been observed. In a case study by Reistad et al. (2016) conjugate aurora was displaced by 3 hours MLT. This was also explained by the combination of moderate IMF $B_Y$ (5 nT) and a large positive tilt angle (17°). This further emphasizes that an highly asymmetric geospace is a common state of the system.

*Data availability.* The data described in this paper are available from the authors on request (nikolai.ostgaard@uib.no)

*Acknowledgements.* This study was supported by the Research Council of Norway under contract 223252/F50 (CoE). S.E.M. was supported by the Science and Technology Facilities Council, UK, grant no ST/N000749/1. The data described in this paper are available from the authors on request (nikolai.ostgaard@uib.no).

We thank S.B. Mende and the IMAGE FUV team at the Space Sciences Laboratory at UC Berkeley for the FUV data. The images were processed using the FUVIEW3 software (http://sprg.ssl.berkeley.edu/image/). We thank Rae Dvorsky and the Polar VIS team at the University of Iowa for the VIS Earth data. The VIS Earth images were downloaded from NASA's Space Physics Data Facility (ftp://cdaweb.gsfc.nasa.gov/pub /data/polar/camera-full/) and processed using the XVIS 2.40 software (http://vis.physics.uiowa. edu/vis/software/).

We thank C. Smith for the ACE magnetic field data and D. McComas for the ACE solar wind data. We acknowledge use of NASA/GSFC's Space Physics Data Facility's OMNIWeb service and OMNI data.

The DMSP SSIES data were downloaded from http://cindispace.utdallas.edu/DMSP/. We gratefully acknowledge the Center for Space Sciences at the University of Texas at Dallas and the US Air Force for providing the DMSP thermal plasma data.

The authors thank the NOAA's National Geophysical Data Center (NGDS) for providing NOAA POES data.

The CHAMP mission was sponsored by the Space Agency of the German Aerospace Center (DLR) through funds of the Federal Ministry of Economics and Technology, following a decision of the German Federal Parliament (grant code 50EE0944). We thank Dr. Patricia Ritter for processing the data.

For the ground magnetometer data we gratefully acknowledge the following: Intermagnet; USGS, Jeffrey J. Love; CARISMA, Ian Mann; CANMOS; the S-RAMP database, K. Yumoto and K. Shiokawa; the SPIDR database; AARI, Oleg Troshichev; the MACCS program, M. Engebretson; Geomagnetism Unit of the Geological Survey of Canada; GIMA; MEASURE, UCLA IGPP, and Florida Institute of Technology; SAMBA, Eftyhia Zesta; 210 Chain, K. Yumoto; SAMNET, Farideh Honary; the institutes that maintain the IMAGE magnetometer array, Eija Tanskanen; PENGUIN; AUTUMN, Martin Conners; DTU Space, Jürgen Matzka; South Pole and McMurdo Magnetometer, Louis J. Lan-





zarotti and Alan T. Weatherwax; ICESTAR; RAPIDMAG; PENGUIn; British Antarctic Survey; McMac, Peter Chi; BGS, Susan Macmillan; Pushkov Institute of Terrestrial Magnetism, Ionosphere and Radio Wave Propagation (IZMIRAN); GFZ, Jürgen Matzka; MFGI, B. Heilig; IGFPAS, J. Reda; University of L'Aquila, M. Vellante; SuperMAG, Jesper W. Gjerloev.

We acknowledge the use of SuperDARN data. SuperDARN is a collection of radars funded by national scientific funding agencies of Aus-

5   tralia, Canada, China, France, Japan, Norway, South Africa, United Kingdom and United States of America. The Virginia Tech SuperDARN database (sftp://sd-data.ece.vt.edu) is automatically accessed by the DaViTpy python toolkit.

Simulation results have been provided by the Community Coordinated Modeling Center at Goddard Space Flight Center through their public Runs on Request system (http://ccmc.gsfc.nasa.gov). The CCMC is a multiagency partnership between NASA, AFMC, AFOSR, AFRL, AFWA, NOAA, NSF, and ONR (William_Longley_112213_4).



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
