# Peer review of "The asymmetric geospace as displayed during the geomagnetic storm on August 17, 2001"

_Annales Geophysicae, 2018_

## Referee Comment (RC1) · Anonymous Referee #1 · 25 Jul 2018

This paper presents a highly interesting study of the dynamical behaviour of the magnetosphere based on asymmetric observations of auroral and convection features in the ionosphere. The combination of a large IMF By component and a large tilt angle due to the season sets up a highly asymmetric system that can be studied in both polar ionospheres in order to probe variations in the internal structure of the magnetosphere. The paper clearly showcases the power of simultaneous auroral image observations in both hemispheres, which can be used to address big questions of magnetospheric physics not possible from in-situ magnetospheric measurements alone. The video produced for the supplementary information provides an excellent way of viewing the event as a whole and of how all the different data sources fit together. Visual information like

this is incredibly useful for helping to understand the strengths and limitations of a data set.

Major Comments:

(1) I have a few questions regarding the determination and portrayal of ionospheric convection in the paper that the authors need to clarify: (i) Does the determination of the convection using the SECs method use any 'in-fill' data (such as a convection climatology e.g., Weimer, Ruohoniemi and Greenwald, Cousins and Shepherd) to cover regions where there are little or no real data? If not, how reliable do you think the convection solutions are in these regions? (ii) The flow vectors shown from the DMSP SSIES instrument appear to be 2-D vectors. Although the cross-track ion velocities measured by the ion drift meter on DMSP SSIES are very reliable data, I had thought that there were issues with the along-track velocity estimates. Is this not the case? (iii) In Figure 11, the convection map in 11C is 'now drawn by hand'. Why? Is it not possible to reproduce the map presented in 11A? (iv) In Figure 11D, the convection map is 'drawn by hand, guided by the few available SuperDARN vectors...'. Is it not possible to use a climatological model to better estimate the global convection here (which SuperDARN or other datasets can provide)?

(2) I also have a query about the scaling between SI13 and VIS. The IMAGE spacecraft is clearly close to apogee for both time intervals considered in the paper (the scaling interval and the two-hemisphere interval). What is the orbit of POLAR at this time? My concern is that at the time of the scaling, POLAR is close to perigee, whereas during the time of the two-hemisphere observations it is closer to apogee. This would mean that the spatial size of the measurement cells in the ionosphere are different for the observations made by VIS in the two hemispheres (this is implied by the video, as the overall field of view of VIS is much smaller in the northern hemisphere when the spacecraft is closer to perigee). Does this difference in VIS cell size in the two hemispheres affect the number of counts that are typically measured within a cell? And does this therefore affect the measured intensity? If so then this might cast some

doubt over the scaling – it may not be as simple as implied in the paper.

Minor Comments:

(1) The authors should consider referencing the work of Grocott et al. in the paper introduction (and elsewhere in the paper if appropriate). For example, Grocott et al. (2010), Superposed epoch analysis of the ionospheric convection evolution during substorms: IMF By dependence, JGR, 115, A00106, doi:10.1029/2010JA015728 (and also see papers referenced within). This paper shows the IMF By-dependent dawn-dusk asymmetry in convection being reduced at substorm onset.

(2) Page 4, Section 2 – At no point is it stated what co-ordinate system is being used for the plotting of the polar data. As IMAGE data are typically in APEX co-ordinates, and SuperDARN and DMSP measurements are often in AACGM co-ordinates, it should be made clear what co-ordinate system has been used in the polar plots.

(3) Page 4, Section 2 – Although there are instrumental references for the IMAGE, POLAR and OMNI data, there are no instrumental references for SuperDARN, SuperMAG, DMSP and CHAMP. This should be rectified.

(4) Page 14, Figure 7 – The blue symbols in panel A are very hard to see on the coloured background. You might want to consider an alternative way of displaying these symbols.

(5) Page 17, Lines 7-10 and Figure 10 – The text implies that the measurements of DeltaMLT combine observations from the poleward edge of the oval with those nearer to the equatorward edge. However, the modelling work presented earlier has already shown that the offsets in these two regions will be different. If the measurements were all from the poleward edge then this would reinforce the suggestion that the changes are substorm related.

(6) Page 19, Figure 11 and Page 20, Figure 12 – How are the reconnection regions (red lines) identified? The equipotential contours in figure 11A suggest that there is flow

across the OCB across most of the nightside (if the dawn convection cell is included).

(7) Page 21, Lines 4-5 and Page 22, Figure 14 – Considering the text 'The equatorward convection just outside the auroral oval near noon is probably an artefact of the inversion, since the solution there is relatively unconstrained by observations compared to the surrounding regions'. This may be true. However, the observed convection variation around noon may also be the signature of split reconnection X-lines. If one considers the preferential anti-parallel reconnection regions on the magnetopause during IMF By-dominated conditions, the reconnection regions are split into two separate regions, one at high latitude in the northern hemisphere, and one at high latitude in the southern hemisphere. This can lead to an ionospheric convection signature that is characterised by two poleward flow regions either side of noon (mapping to the two reconnection regions) with a reduced flow region, and possibly no flow across the OCB, very close to noon. (See Chisham et al. (2002), Ionospheric signatures of split reconnection X-lines during conditions of IMF Bz<0 and |By|~|Bz|: Evidence for the antiparallel merging hypothesis, JGR, 107(A10), 1323, doi:10.1029/2001JA009124).

(8) Page 25, Line 1 – 'Mapped into the plasma sheet they cover huge regions'. The use of the word 'huge' is very vague. It should be possible to put some approximate value to this.

Other Comments and Typos:

(1) Page 3, Line 12 – 'do' should be 'does'.

(2) Page 3, Line 24 – 'is' should be 'are'.

(3) Page 4, Line 20 – 'is' should be 'are'.

(4) Page 5, Line 26 – It would read better if '. . .more energetic electrons drifting towards dawn scattered into the loss cone. . .' was written '. . .more energetic electrons drifting towards dawn where they are scattered into the loss cone. . .'.

(5) Page 7, Line 4 – 'weigh' should be 'weight'.

(6) Page 15, Line 9 – 'Figure 7B and 7B' should be 'Figures 7B and 7C'.

(7) Page 15, Section 4.5 title – This should possibly be retitled 'Why latitudinally wider aurora in the southern dawn?' to distinguish it from being wider in MLT.

(8) Page 18, Line 3 – 'have' should be 'has'.

(9) Page 18, Line 9 – '. . .for the period we observe large asymmetry' should be '. . .for the period when we observe large asymmetry'.

(10) Page 20, Line 3 – remove one of the 'the's.

(11) Page 20, Line 3 – 'hemisphere' should be 'hemispheres'.

(12) Page 23, Line 12 – remove 'being'.

(13) Page 23, Line 14 – 'encounters' should be 'encounter'.

(14) Page 23, Line 19 – 'leads' should be 'lead'.

(15) Page 25, Line 20 – 'summerize' should be 'summerized'.

(16) Page 25, Line 28 – 'in' should be 'at'.

(17) Page 27, Line 5 – The SuperDARN acknowledgement should include 'Italy' in the list of countries.

---

## Referee Comment (RC2) · Anonymous Referee #2 · 3 Aug 2018

Comment on "The asymmetric geospace as displayed during the geomagnetic storm on August 17, 2001" by Nikolai Ostgaard et al.

The paper contains a thorough analysis of magnetospheric asymmetry, as it reveals in the auroral oval morphology. However, I would suggest some significant changes in presentation style

1. It is very hard (for a not fully motivated reader) to follow all presented details in aurora images, starting from Fig. 3 and 4 and later in explanations. I would suggest to mark and number all features, which are used to prove asymmetry, and summarize them in the table. All following discussion and figures should refer to those numbers.

[Figure]

2. The paper is overloaded by description of caveats, complications, alternative explanations of observed features. However practically none of these details actually affects conclusions, which are based on very simple observation signatures. Probably these details are necessary to prove validity of analysis, but they should not distract readers from the main line.

3. Discussion of Fig 12 is very unclear. Are there any statistically significant changes of OCB in it ?

4. page 20 line 10. GREEN LINE is where?

5. line 20 and on page 21. Mark clearly all mentioned flows in Figure 14.

---

## Short Comment (SC1) · 7 Aug 2018

The comments summarise a discussion of this manuscript by the Space Plasma Physics Group at the Mullard Space Science Lab, UCL.

This manuscript aims to study the effect of the By component of the IMF on the Earth's magnetosphere, in particular the asymmetries observed in the magnetosphere during periods of high IMF By. The manuscript analyses a case study of a geomagnetic storm which occurred in August 2001 during which the magnetosphere was exposed to a strong IMF By component. To study this particular storm, many available data sources are utilised to build up a picture of the asymmetries observed in the magnetosphere.

[Figure]

Comments:

Two of these data sources IMAGE (SI13) and POLAR (VIS) are used to explore conjugacy in auroral features of the two hemispheres. In Figures 3 and 4, which present a comparison between the northern and southern auroral ovals, we suggest that the scales be converted into units of energy flux or Rayleighs rather than using counts. This may help to better illustrate the conjugacy in the auroral features. In addition, we noted that the displacement of the conjugacy observed in Figure 3A is difficult to ascertain, given the reduced field of view in the VIS data.

In Figure 11 we feel that there are too few data points used in the calculation of the convection patterns and thus does not provide strong enough evidence to conclude that reconnection is occurring in the northern lobe region.

In the discussion sections 4.4, 4.5 and 4.6 we feel that the clarity of the explanations provided could be improved through the use of schematic diagrams of the magnetospheric processes which are described. Figure 8 is a helpful illustration but could be described in more detail within the main text.

The description in section 4.6, might take into consideration the time scales involved for the effects of IMF By to propagate through the lobes, into the plasma sheet and create the asymmetries in the magnetosphere. This would help to place this discussion in the context of the time scales of By being induced and transported in the closed magnetosphere from the literature referenced in the introduction section (for example, Tenfjord et al. (2015, 2018), Motoba et al. (2010), Rong et al. (2015)). An additional reference which may be useful for this discussion or the introduction is Timescales for the penetration of IMF By into the Earth's magnetotail Browett et al. (2016).

We also suggest that more discussion on how a By component is induced in the magnetosphere could be included in the introduction section, especially given that one of the main findings presented in the manuscript provides evidence for tail reconnection reducing the induced By component in the magnetosphere.

A few smaller points that we note are that more detailed description might be added to the figure captions and the title of the manuscript might be considered for revising, in particular the term 'asymmetric geospace'.

As noted, many data sources are used in the manuscript to study this particular storm event. We suggest that another helpful data source might be the Cluster spacecraft which should be located in the magnetotail during this period and may provide direct measurements of the By component in the plasma sheet.

Overall, the manuscript is presented well and written in a logical order. We hope our feedback is helpful in the development of this manuscript.

---

## Author Response (AR1)

Response to referee #1
* * *
First, we will thank the referee for carefully reading our manuscript and for providing very useful comments. We will respond to the comments one by one, and also indicate the changes (how and where) they have been implemented in the marked-up version (see red text below)

Major comments:
——————————

1) Determination of convection pattern.
(i) We are not using any «in-fill» data to cover regions with little data, but we take advantage of the SuperMAG data that can be used in the sunlit part, and as can be seen the green dots (SuperDARN) and brown arrows (SuperMAG) gives us quite good coverage, see also Figure 14. Several places we emphasize that the patterns are derived «entirely from data». To make it even more clear, we  now use «measurements» instead of «data» (page 1 line 13, page 3 line 29, page 13 line 2, page 19 line 15)
(ii) It is true that sometimes DMSP data show very large values which can not be trusted, but this is not the case here. We are only using validated data as input for the convection model.
(iii) Yes, you are right, it is not needed to draw Figure 11C by hand. We have fixed this.
(iv) SuperDARN convection map for the Southern Hemisphere  is available on their web-page and supports our regions of dayside and nightside reconnection in the Southern Hemisphere. However, there are no data for the entire dawn cell and the superDARN map of that cell is not consistent with the auroral imaging data in the dawn. We have added (page 23, line 22): «We should mention that there exists a SuperDARN convection map (see their web page), which supports the locations of dayside and tail reconnection. However, there are no data in the dawnside and the convection reversal in this map is not consistent with the poleward edge of the auroral in this hemisphere.»

2) Scaling between WIC and SI13. As a general rule, as long as the auroral features are larger than the pixel size,  the area covered by a pixel increases by $r^2$ whereas the luminosity from the source decreases by $r^2$, which means that these two effects cancel. Having said that, we are not claiming that we have the scaling correct, and we emphasize several places that we are not comparing absolute intensities, but only shapes and relative intensity increases.
See Page 3 line 18 and page 6 line 8.

Minor comments
——————————
1) Thanks for pointing out the Grocott paper. We have referenced this result both in the introduction (page 3 line 34 and page 19 line 1)

2) In the Data section we have specified the different coordinate systems that has

been used: APEX for IMAGE and Polar and AACGM for SuperDARN and DMSP. We also point out that the difference between these coordinate systems is negligible for the results presented in this paper. See page 4 line 15 and 20

3) We have added instrumental references for SuperDARN, SuperMAG, DMSP and CHAMP. (See page 4 line 18)

4) Symbols in Figure 7. We have tried without success to find a different color that would display better, so we keep it blue (a color that is not in color scale for the aurora). Then they are consistent with the blue in panel B and C.
We have added in the figure caption: «The blue symbols are at the same locations in all panels»

5) Yes it is true that modeling indicates larger asymmetry at the poleward edge than at the equator ward edge. in Figure 3A one can see that the features marked 1 and 2 is bent dawnward at the poleward edge. However, it is not possible to determine from SI13 how bent they are in the north. So we decide to keep it as is. The main point here is that the asymmetries are only 0.5-1 hour MLT.

6) In Figure 11A OCB is identified by the poleward edge of the aurora. The colors in Figure 12 only indicate time and the purpose is to show that the OCB does not move, which tells us that flux is indeed transferred across the OCB.
It is correct that there may be reconnection also beyond 22.5 MLT in the north, but the contours indicate a much weaker flow. We have added that 18-22 MLT is where the most intense flows are observed, and that there are weaker flows across the OCB dawnward of this. (Page 19, line 20)

7) This is an excellent comment, and we also believe that we have a split reconnection line. This is what one would expect if the reconnection site at the magnetopause in the Southern Hemisphere is also included. This location would have its magnetic footprint around 11 MLT in the northern hemisphere. We have made a comment about that. «Since the flow between 11 MLT and 13 MLT is mostly along the OCB, the flows seen within the green and blue circle indicate split reconnection X-lines, similar to what was reported by Chisham et al., 2002 during similar IMF conditions.» (Page 21 line 15 - page 23 line 1)

8) Instead of huge regions we now state: «Mapped into the plasma sheet (not shown) they cover huge regions (dY: 4-5 Re and dX: 10 Re» (page 25 line 8)

The comments and typos
——————

1-17: thanks and they are all changed according to the suggestions

Response to referee #2
* * *
First, we will like to thank the referee for carefully reading our manuscript and for providing very useful comments.

We will respond to the comments one by one, and also indicate the changes we will make in the manuscript. At this point we are asked to only provide response, not a marked-up new version, so we will point to where in the original version we suggest to make the changes

1. In Figure 3 and 4 we have already encircled the auroral features we claim to be conjugate. In addition we have marked two other features in Figure 3A with numbers. To accommodate the reviewers suggestion, we have added a table summarizing the asymmetries from Figure 3 and 4 at the end of Section 4.2. These are the deltaMLTs that are used in Figure 10 as well.

2. As there are many people in the community that would be skeptical to such large asymmetries as we have identified here, we have decided to show supporting data for our interpretation and to also explore alternative interpretations.
As you correctly state, we do not find sufficient support for these alternative interpretations, and consequently they do not change the conclusions, but rather strengthen them. For this reason we prefer to keep the manuscript as is in this regard.

3. Discussion of Figure 12. This figure is included to show that the OCB in the north does NOT move in the two sectors where we claim to have dayside (11-15 MLT) and tail reconnection (18-22 MLT), which means that the flows in these regions (Figure 11A) are indeed across the OCB, which means that these are reconnection locations.
To make this clear we have added a sentence (page 19, line 21): «To check whether the flow pattern between 11-15 MLT and 18-22 MLT seen in Figure 11A is really flows across the OCB and not only a motion of the OCB itself, we show, in Figure 12, the time evolution ……..»

4. On page 19 line 29, we have added a reference to Figure 11A: «In Figure 11A, we have marked lobe reconnection by a green line from 15-17.5 MLT»

5. We have marked the different regions of flows in Figure 14 and made proper references to these in the text.
- Page 21, line 9: «(75 deg, make with a red circle)»
- Page 21, line 11: «(Figure 14, marked with a blue circle)
- Page 21, line 14: «Flow across the OCB is also seen at 11 MLT and <70 (green circle)

Response to M. Mooney:
* * *
First we would like to thank you and your group for taking time and effort to read and discuss this manuscript.

Here is our response:

1. Energy flux or Rayleigh versus counts. To derive energy flux from images is not a straightforward thing to do and require detailed modeling. Frey et al., 2003 have given us tools to do this for IMAGE, but there are no such tools for VIS camera. Converting to Rayleigh is just a multiplication of a constant which gives the impression of comparing similar images, but they are not. The cameras measure different emissions and different wavelength bands. We believe it is most honest to use counts and describe how we (as best we can) scale the images.

2. FOV of VIS in Figure 3A. We agree and this is also pointed out (page 9 line 8) and uncertainty is included in Figure 10

3. Our main argument for having lobe reconnection in the North is the auroral lobe spot seen at very high latitudes (Figure 13). In addition, Figure 14 shows sunward convection at high latitudes (now marked with red circle to clarify) in the same region as we see the spot. There are both green dots (SuperDARN line-of-sight) and brown arrows (SuperMAG) in this region.

4. Instead of expanding the text (in 4.4, 4.5 and 4.6) in an already long article we have repeated a reference to Tenfjord et al., 2015, where it is explained in great detail (both theoretically and by MHD modeling results) how By is induced by the lobe pressure. See page 16 line 5.

5. Time scales that are involved. We have explained the contradicting results about time scales for establishing an induced By component in the introduction (see page 2), and now we also included Browett et al. (2016) paper as another paper claiming 2 hours delay, or more (Page 2 line 22). Since we do not have any IMF By-polarity changes during this magnetic storm, we cannot address how fast By is induced. However, as we point out on page 2, this has been shown by two papers by Tenfjord et al., (2017 and 2018). In the present paper we do think we have sufficient data to support that asymmetry is reduced by substorms due to increased reconnection. This is important and contradicts the idea that reconnection is the process by which asymmetry is introduced. Our group has just submitted a paper (to JGR) which addresses also the time scales involved in removing asymmetry due to increased reconnection. This paper by Ohma et al. will be published soon.

6. We believe the term asymmetric geospace is rather accurate, because there is asymmetry in both reconnection locations on the magnetopause (dayside) - asymmetric magnetic pressure in the lobes, which creates asymmetric footprints of field lines and aurora, and there are asymmetric convection patterns in the two ionospheres. This means that all main regions of geospace are asymmetric.

7. Yes, Cluster was in the magnetosphere during this time and could have been used (see Echer et al., JGR113, A05209, doi:10.1029/2007JA012624). However, this is already a long paper with a lot of data, and we believe we have sufficient support for our main conclusions.

Ann. Geophys. Discuss.,
https://doi.org/10.5194/angeo-2018-65-AC1, 2018

[Figure]

First, we will thank the referee for carefully reading our manuscript and for providing very useful comments. We will respond to the comments one by one, and also indicate the changes (how and where) we will make to the manuscript. At this point we are asked to only provide response, not a marked-up new version, so we will point to where in the original version we suggest to make the changes

[Figure]

**1 Major comments**

1) Determination of convection pattern.

(i) We are not using any "in-fill" data to cover regions with little data, but we take advantage of the SuperMAG data that can be used in the sunlit part, and as can be seen the green dots (SuperDARN) and brown arrows (SuperMAG) gives us quite good coverage, see also Figure 14. Several places we emphasize that the patterns are derived "entirely from data". To make it even more clear, we suggest to use "entirely from measurements" instead of "entirely from data" when we refer to the model. We will also make explicitly statement: "To estimate the global plasma flow pattern (in a coroating frame), we adopt a novel purely data based multi-instrument approach, without using any empirical model to fill in regions with data gaps."

(ii) It is true that sometimes DMSP data show very large values which can not be trusted, but this is not the case here. We are only using validated data as input for the convection model. The SSIES data are provided with quality flags in both directions (along and cross track). We assign a common quality flag to each vector which is equal to the poorest quality component. We then assign a weight to the data point which depends on this quality flag, down-weighting poor quality data points in the final inversion.

(iii) Yes, you are right, it is not needed to draw Figure 11C by hand. We will fix that.

(iv) SuperDARN convection map for the Southern Hemisphere is available on their webpage and supports our regions of dayside and nightside reconnection in the Southern Hemisphere. However, there are no data for the entire dawn cell and the superDARN map of that cell is not consistent with the auroral imaging data in the dawn. We suggest to add in Section 4.2.7: "We should mention that there exists a SuperDARN convection map (see their web page), which supports the locations of dayside and tail reconnection. However, there are no data in the dawnside and the convection reversal in this

map is not consistent with the poleward edge of the auroral in this hemisphere."

2) Scaling between WIC and SI13. As a general rule, as long as the auroral features are larger than the pixel size, the area covered by a pixel increases by $r^2$ whereas the luminosity from the source decreases by $r^2$, which means that these two effects cancel. Having said that, we are not claiming that we have the scaling correct, and we have emphasized several places that we are not comparing absolute intensities, but only shapes and relative intensity increases. (see Page 3 line 17 and page 6 line 1).

**2  Minor comments**

1) Thanks for pointing out the Grocott paper. We will reference this result both in the introduction:

"Reduction of the BY dependent dawn-dusk asymmetry in convection pattern after substorm onset has also been reported by Grocott et al. (2010)"

and at the end of Section 4.6:

"Reduction of IMF BY related asymmetries during substorm expansion phase has been observed both in conjugate auroral images (Østgaard et al., 2011a) and in convection patterns (Grocott et al., 2010)."

2) In the Data section we will specify the different coordinate systems that has been used: APEX for IMAGE, Polar and DMSP, and AACGM for SuperDARN. We also point out that the difference between these coordinate systems is negligible for the results presented in this paper. Two sentences will be added:

"For all the imaging data presented in this paper APEX coordinates are used."

"For SuperMAG and DMSP we have used APEX coordinates, while for SuperDARN AACGM coordinates are used. The APEX and AACGM coordinate systems are almost

identical (Laundal and Richmond, 2016) at high latitudes and is negligible for the results presented in this paper."

3) We will add instrumental references for SuperDARN, SuperMAG, DMSP and CHAMP:

SuperDARN (Greenwald et al., 1995) , SuperMAG (Gjerloev, 2012) DMSP (Rich and Hairston, 1994) CHAMP (Reigber et al., 2002)

4) Symbols in Figure 7. We have tried without success to find a different color that would display better, so we keep it blue (a color that is not in color scale for the aurora). Then they are consistent with the blue in panel B and C. We have added in the figure caption: "The blue symbols are at the same locations in all panels"

5) Yes it is true that modeling indicates larger asymmetry at the poleward edge than at the equator ward edge. in Figure 3A one can see that the features marked 1 and 2 is bent dawnward at the poleward edge. However, it is not possible to determine from SI13 how bent they are in the north. So we decide to keep it as is. The main point here is that the asymmetries are only 0.5-1 hour MLT.

6) In Figure 11A OCB is identified by the poleward edge of the aurora. The colors in Figure 12 only indicate time and the purpose is to show that the OCB does not move, which tells us that flux is indeed transferred across the OCB. It is correct that there may be reconnection also beyond 22.5 MLT in the north, but the contours indicate a much weaker flow. We will add that 18-22 MLT is where the most intense flows are observed, and that there are weaker flows across the OCB dawnward of this.

7) This is an excellent comment, and we also believe that we have a split reconnection line. This is what one would expect if the reconnection site at the magnetopause in the Southern Hemisphere is also included. This location would have its magnetic footprint around 11 MLT in the northern hemisphere. We will make a comment about that. "Since the flow between 11 MLT and 13 MLT is mostly along the OCB, the flows seen

within the green and blue circle indicate split reconnection X-lines, similar to what was reported by Chisham et al., 2002 during similar IMF conditions." NB: In response to the other reviewer, green and blue circles have been added in Figure 14 to point out where the flow across the OCB on the dayside are observed.

8) Instead of huge regions we will state: "Mapped into the plasma sheet (not shown) they cover huge regions ($\triangle$Y: 4-5 Re and $\triangle$X: 10 Re)"

**3   Other comments and typos**

1-17: thanks and they will all be changed according to the suggestions

[Figure]

Ann. Geophys. Discuss.,
https://doi.org/10.5194/angeo-2018-65-AC2, 2018

[Figure]

First, we will like to thank the referee for carefully reading our manuscript and for providing very useful comments. We will respond to the comments one by one, and also indicate the changes we will make in the manuscript. At this point we are asked to only provide response, not a marked-up new version, so we will point to where in the original version we suggest to make the changes.

1. In Figure 3 and 4 we have already encircled the auroral features we claim to be conjugate. In addition we have marked two other features in Figure 3A with numbers. To accommodate the reviewers suggestion, we will add a table summarizing the asym-

metries from Figure 3 and 4 at the end of Section 4.2. These are the ΔMLTs shown by diamonds in Figure 10 as well.

2. As there are many people in the community that would be skeptical to such large asymmetries as we have identified here, we have decided to show supporting data for our interpretation and to also explore alternative interpretations. As you correctly state, we do not find sufficient support for these alternative interpretations, and consequently they do not change the conclusions, but rather strengthen them. For this reason we prefer to keep the manuscript as is in this regard.

3. Discussion of Figure 12. This figure is included to show that the OCB in the north does NOT move in the two sectors where we claim to have dayside (11-15 MLT) and tail reconnection (18-22 MLT), which means that the flows in these regions (Figure 11A) are indeed across the OCB, which means that these are reconnection locations. To make this clear we will add a sentence

"To check whether the flow pattern between 11-15 MLT and 18-22 MLT seen in Figure 11A is really flows across the OCB and not only a motion of the OCB itself, we show, in Figure 12, the time evolution . . . .. .. ."

4. Green line: we will add a reference to Figure 11A: "In Figure 11A, we have marked lobe reconnection by a green line from 15-17.5 MLT"

5. We have marked the different regions of flows in Figure 14 and will make proper references to these in the text. The text will then read:

"The derived convection shown by blue arrows in Figure 14 indicates sunward flow at very high latitudes ($75°$, marked with a red circle), just where we see the spot and the upward field-aligned current from CHAMP. The LFM model also predicts a strong upward current on open field lines between $70°$ and $80°$ at 17-18 MLT (not shown). Flow across the OCB (Figure 14, marked with a blue circle) indicating dayside reconnection is seen around 15 MLT and $< 70°$ with downward current (CHAMP) and proton precip-

itation (SI12 image). Flow across the OCB is also seen at 11 MLT and $< 70°$ (green circle) consistent with the dayside reconnection region we have indicated by the red line in Figure 11A. "

We have uploaded a revised figure 14.

[Figure]

SI12

SI13

500.0 nT, 1000 m/s

WIC

CHAMP

18

06

00

**Fig. 1.** Revised Figure 14

[Figure]